# Interfaces between hexagonal and cubic oxides and their structure alternatives

Hua Zhou[1,2], Lijun Wu[3], Hui-Qiong Wang [1,4], Jin-Cheng Zheng [1,4], Lihua Zhang[5], Kim Kisslinger[5], Yaping Li [1], Zhiqiang Wang[1], Hao Cheng[1], Shanming Ke[2], Yu Li[2], Junyong Kang[1] & Yimei Zhu[3]

Multi-layer structure of functional materials often involves the integration of different crystalline phases. The film growth orientation thus frequently exhibits a transformation, owing to multiple possibilities caused by incompatible in-plane structural symmetry. Nevertheless, the detailed mechanism of the transformation has not yet been fully explored. Here we thoroughly probe the heteroepitaxially grown hexagonal zinc oxide (ZnO) films on cubic (001)-magnesium oxide (MgO) substrates using advanced scanning transition electron microscopy, X-ray diffraction and first principles calculations, revealing two distinct interface models of (001) ZnO/(001) MgO and (100) ZnO/(001) MgO. We have found that the structure alternatives are controlled thermodynamically by the nucleation, while kinetically by the enhanced Zn adsorption and O diffusion upon the phase transformation. This work not only provides a guideline for the interface fabrication with distinct crystalline phases but also shows how polar and non-polar hexagonal ZnO films might be manipulated on the same cubic substrate.

[1] Fujian Provincial Key Laboratory of Semiconductors and Applications, Collaborative Innovation Center for Optoelectronic Semiconductors and Efficient Devices, Department of Physics, Xiamen University, Xiamen 361005, China. [2] College of Materials Science and Engineering and Shenzhen Key Laboratory of Special Functional Materials, Shenzhen University, Shenzhen 518060, China. [3] Condensed Matter Physics and Materials Science Department, Brookhaven National Laboratory, Upton, NY 11973, USA. [4] Xiamen University Malaysia, Sepang 43900, Malaysia. [5] Center for Functional Nanomaterials, Brookhaven National Laboratory, Upton, NY 11973, USA. Hua Zhou and Lijun Wu contributed equally to this work. Correspondence and requests for materials should be addressed to H.-Q.W. (email: hqwang@xmu.edu.cn) or to J.-C.Z. (email: jczheng@xmu.edu.cn) or to Y.Z. (email: zhu@bnl.gov)

Recently, in order to fabricate multifunctional electronic devices, researchers have focused on the study of interfacial structures between two materials with distinct symmetry groups[1–6], for example, one with an $Fm$-3m (e.g. MgO) or $Pm$3m (e.g., SrTiO$_3$) space group and the other with a $P6_3mc$ space group (e.g., ZnO). However, some issues may arise due to the change of growth directions depending on growth conditions. For example, our group[7–9] and others[10–14] have demonstrated that the growth direction (either polar (001) or non-polar (100)) of wurtzite ZnO thin films strongly depends on the substrate temperature and growth pressure. Remarkably, it has recently been observed that a change in the orientation of the coupling plane with substrates can improve device properties due to the removal of central symmetry for ZnO. This effect is similar to the phenomenon where different planes perform different catalytic activity for the same catalyst[15,16]. For example, non-polar m-plane ZnO instead of c-plane coupling with ZnMgO or other semiconductors can reduce the Stark effect[17–19]. In addition, it is beneficial to prepare high-quality films to improve the properties of the optoelectronic devices[20,21]. Thus, the design of coupling with different planes to improve the device properties becomes a common method for the study of ZnO. Naturally, it is of great importance to address the issues of growth orientation and microscopic structures of the interface between different planes for the fabrications and applications of novel devices by synthesizing ZnO with other materials. Nevertheless, there are still lack of enough literatures to reveal the features of the microscopic structure of the interface for the different growth orientations and even less to explain the related mechanism.

In this work, firstly, we obtained a diagram of the growth orientations transformation of the ZnO films between the c-plane and m-plane prepared with different growth temperatures or oxygen pressures using molecular beam epitaxy. Secondly, the microscopic structures of the interface between the ZnO films with wurtzite structure and the MgO (001) substrate with cubic structure were revealed through the scanning transmission electron microscopy (STEM) with high-angle annular dark-field (HAADF) detector and X-ray diffraction (XRD) $\phi$-scan. Finally, according to the classical nucleation theory, we propose that the physical essence of the orientation transformation (or alternative) originates from the difference of the nucleation process. The involved interface energies, surface energies, and energy change of transformation were obtained through first principles calculations based on the density functional theory (DFT) using the Vienna ab initio simulation package (VASP) code[22,23]. Our theoretical calculation results show that, under the condition that the contact angle of c-plane ZnO is much larger than m-plane, the nucleation barrier of the [001]$_{ZnO}$ orientation could be smaller than that of the [210]$_{ZnO}$ orientation. On the other hand, the higher growth temperature will induce the atom diffusion and then enhance the Zn atom adsorption and O atom diffusion on the substrate, which stimulate the transformation of growth orientation. This work not only explains the previous experimental results of the growth orientations transformation of the ZnO films or other materials under different growth conditions, but also paves the way to integrate optical materials of wurtzite structure (such as ZnO, GaN, etc) with multi-ferromagnetic and/or high dielectric materials of perovskite structure (such as SrTiO$_3$, BaTiO$_3$, etc).

## Results

### The diagram of growth orientation transformation.
XRD results and in-situ reflection high energy electron diffraction (RHEED) patterns (Supplementary Fig. 1 and Supplementary Note 1) from the ZnO films demonstrate the growth orientation transformation of the film from c-plane to m-plane with the

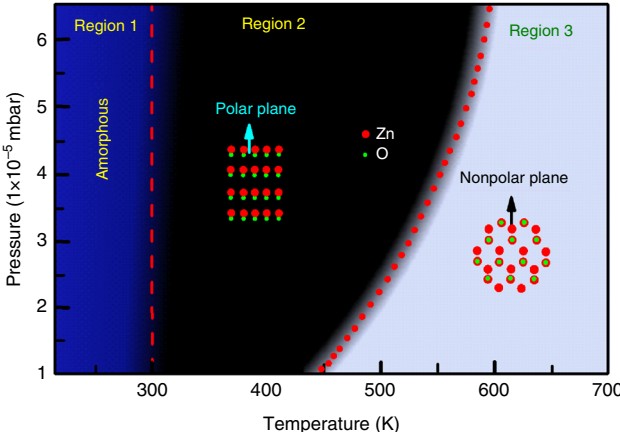

**Fig. 1** Diagram of the ZnO growth orientations. The transformation of the growth orientation is tailored by the growth temperature and O$_2$ partial pressure. It is similar to a conventional phase transition from water to water-vapor[22,23] or a pressure-temperature relationship diagram of the reaction CaMg(CO$_3$)$_2$ (dolomite) + 2SiO$_2$ (coesite) = CaMgSi$_2$O$_6$ (diopside) + 2CO$_2$ (vapor)[24]

change of growth temperature and O$_2$ pressure. A phase diagram of growth orientation as a function of temperature and O$_2$ pressures can thus be obtained, as shown in Fig. 1. When the ZnO film growth occurs with the conditions in region 2 (black) or region 3 (gray), the corresponding growth orientations of the ZnO films grown on (001)$_{MgO}$ substrates are along [001]$_{ZnO}$ or [100]*$_{ZnO}$ azimuth orientations, respectively (Subscripts MgO and ZnO represent MgO substrate and ZnO film, respectively. Superscript "*" indicates the index based on reciprocal lattice). Note, [100]*$_{ZnO}$ is the normal of (100)$_{ZnO}$ plane, and parallel to [210]$_{ZnO}$ direction. When the growth temperature is below about 300 K (region 1 (blue)), the ZnO thin film is of amorphous structure. Remarkably, this diagram of the growth direction transformation from [001]$_{ZnO}$ direction to [210]$_{ZnO}$ direction (as illustrated by the dotted line) is akin to a conventional phase transition from water to water-vapor[24,25] or a pressure-temperature relationship diagram of the reaction CaMg(CO$_3$)$_2$ (dolomite) + 2SiO$_2$ (coesite) = CaMgSi$_2$O$_6$ (diopside) + 2CO$_2$ (vapor)[26].

### The relationship of the interfaces.
We study the relationships between the two types of the interfaces that couple the c- and m-planes, respectively, with the same MgO substrates using STEM, electron diffraction (ED) and XRD $\phi$-scan. Figure 2 shows the STEM images acquired by HAADF detector from the c-ZnO film with the beam along the MgO [110] direction. The bright contrast dots on the left side and the weak contrast dots on the right side correspond to Zn and Mg columns, respectively, since the intensity of the STEM-HAADF image is approximately proportional to Z$^{1.7}$ (Z: atomic number). There are two kinds of c-ZnO domains. One shows a centered-rectangle like atomic arrangement, which is consistent with the ZnO [100] projection (Fig. 2a). The other has a close-packed atomic arrangement corresponding to a ZnO [−1−20] projection (Fig. 2b). From these images, we can reconstruct the interface structure between ZnO film and MgO substrate. Figure 2c shows the ZnO [100] (top) and [−1−20] (bottom) projection along with MgO [110] projection, respectively, for c-ZnO domains I and II. Figure 2d shows the top view with one ZnO (001) layer on one MgO (001) layer (marked by an orange rectangle in Fig. 2c) viewed along the film normal. These

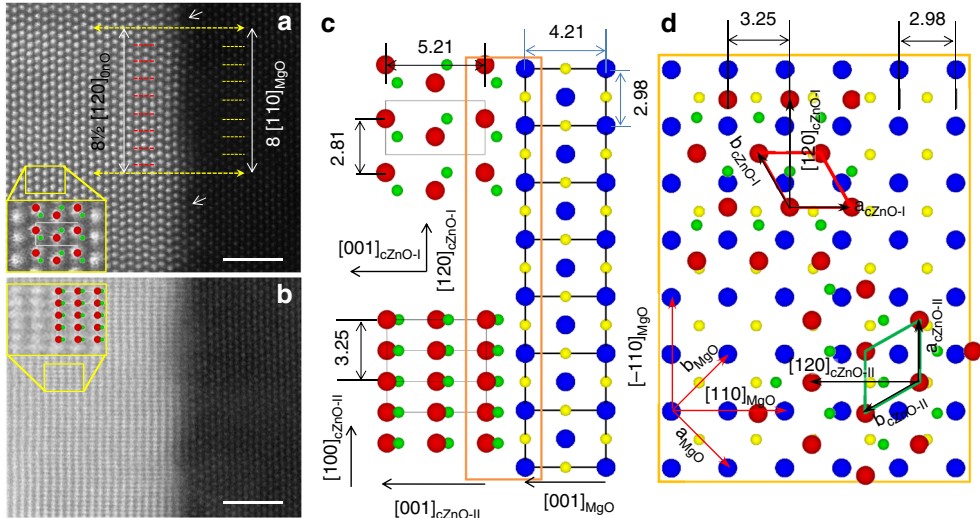

**Fig. 2** Orientation relationship for c-ZnO. Cross-section scanning transmission electron microscopy (STEM)-high-angle annual dark-field (HAADF) images of (**a**) c-ZnO-I and (**b**) c-ZnO-II domains of polar film. Scale bar, 2 nm. The beam is parallel to the $[110]_{MgO}$ direction. The high contrast dots on the left correspond to Zn atoms, while weak contrast dots in the right correspond to Mg atoms. The O atom is invisible due to its low atomic number. The insets show the magnified images from the area outlined by the yellow rectangles with the atomic projection of $[100]_{cZnO-I}$ in (**a**) and $[-1-20]_{cZnO-II}$ in (**b**) embedded (red: Zn, green: $O_{epi}$, yellow: $O_{sub}$, blue: Mg). $[100]_{cZnO-I}$ is aligned with $[110]_{MgO}$, showing centered-rectangle atomic arrangement. The c-ZnO-II is rotated 90° along [001] direction in terms of ZnO-I and viewed along $[-1-20]_{cZnO-II}$, showing close-packed rectangular atomic arrangement. The two yellow dashed arrows in **a** indicate the alignment of a ZnO row with respect to an MgO row. **c** Projection of the interface for c-ZnO-I along [100] direction (upper panel) and c-ZnO-II along [−1−20] direction (lower panel) derived from STEM images. **d** Top view (one layer of $(001)_{ZnO}$ plane on top of one layer of $(001)_{MgO}$ plane) of the interface, which is the interface area outlined by the brown rectangle in **c** (rotated 90° clockwise). The unit cell of the c-ZnO-II domain (outlined by green parallelogram) is rotated 90° from that of the c-ZnO-I domain outlined by red parallelogram

STEM results reveal the interface relationships as follows:

$$[001]_{cZnO-I} \| [001]_{MgO}, [120]_{cZnO-I} \| [-110]_{MgO}, [100]_{cZnO-I} \| [110]_{MgO}; \quad (1)$$

and

$$[001]_{cZnO-II} \| [001]_{MgO}, [100]_{cZnO-II} \| [-110]_{MgO}, [-1-20]_{cZnO-II} \| [110]_{MgO}, \quad (2)$$

where subscripts cZnO-I and cZnO-II represent c-ZnO domain I and domain II, respectively. A three-dimensional view and the related description are shown in Supplementary Fig. 2 and Supplementary Note 2, respectively. To verify the above relationships, we performed XRD φ-scan experiments, as shown in Fig. 3a. There are twelve {011} peaks over the 0–360° scan, indicating two c-ZnO domains. The alignment of $011_{cZnO-I}$ peak with $-111_{MgO}$ peak indicates the out-of-plane component $[001]_{cZnO-I} \| [001]_{MgO}$, and the in-plane component $[010]^*_{cZnO-I} \| [-110]_{MgO}$. Because of $[010]^*_{cZnO-I} \| [120]_{cZnO-I}$, we obtain $[120]_{cZnO-I} \| [-110]_{MgO}$, which agrees with the result obtained from STEM imaging. Similarly, we confirm the interface relationship for the c-ZnO domain II. The above orientations are further confirmed by electron diffraction pattern (EDP), as shown in Fig. 3b, where the [100] zone of ZnO domain I and the [−1−20] zone of ZnO domain II are aligned with the [110] zone of the MgO substrate.

For m-ZnO film (ZnO (100) grown on MgO (001) plane), there are four pairs of ZnO (010) peaks over the 0–360° scan, indicating that there are four m-ZnO domains (Fig. 3c). One pair of 010 peaks, which are 180° apart (labeled in black text $010_{mZnO-I}$ and $1-10_{mZnO-I}$ with subscript mZnO-I representing m-ZnO domain I), deviate 301° (or 360°−301° = 59°) and 121°, respectively, from the $111_{MgO}$ peak. This indicates that their in-plane components $[-120]^*_{mZnO-I}$ (or $[010]_{mZnO-I}$ in real-lattice) and

$[1-20]^*_{mZnO-I}$ (or $[0-10]_{mZnO-I}$) deviate 59° and 121°, respectively, from the in-plane component $[110]_{MgO}$ of MgO. Because the angle between MgO [4−10] and [110] is 59°, we deduce that $[010]_{mZnO-I}$ is parallel to $[4-10]_{MgO}$. Moreover, as the angle between ZnO [011] and [010] is about 58°, $[011]_{mZnO-I}$ is roughly parallel to $[110]_{MgO}$ (just 1° difference). Therefore, when the sample is tilted to the $[110]_{MgO}$ zone, the $[011]_{mZnO-I}$ zone should be observed, and is shown in Fig. 3d using EDP taken from this domain. The above orientation relationship is straightforward in our STEM-HAADF observations, as shown in Fig. 4. The first one shows a center-rectangle pattern with dumbbell-like dots, corresponding to the ZnO [011] projection (Fig. 4a). The STEM-HAADF image calculated based on the multislice method (inset with white outline in Fig. 4a) agrees with the observation very well, confirming the orientation of ZnO [011]. Although the Mg atoms in the MgO substrate (right in Fig. 4a) are well resolved, their shape is slightly elongated vertically, indicating that the MgO substrate slightly deviates from the [110] zone. A simulation with MgO deviating from the [110] zone by 1° (embedded in Fig. 4a, outlined by green lines) agrees with the experimental image very well. When ZnO [011] deviates from MgO [110] by 1°, its [010] is parallel to $[4-10]_{MgO}$, which is consistent with the XRD φ-scan result. On the basis of this, we can reconstruct the interface structure between the m-ZnO film and the MgO substrate, as shown in the top of Fig. 4c (ZnO [011] projection along with MgO [110] projection) and Fig. 4d (viewed along the film normal with one layer of ZnO (100) on one layer of $(001)_{MgO}$). A 3D view is also shown in Supplementary Fig. 2. The orientation relationship is as follows:

$$(100)_{mZnO-I} \| (001)_{MgO}, [010]_{mZnO-I} \| [4-10]_{MgO}, [001]_{mZnO-I} \| [140]_{MgO}. \quad (3)$$

Another pair of (010) peaks in Fig. 3c (labeled as black text

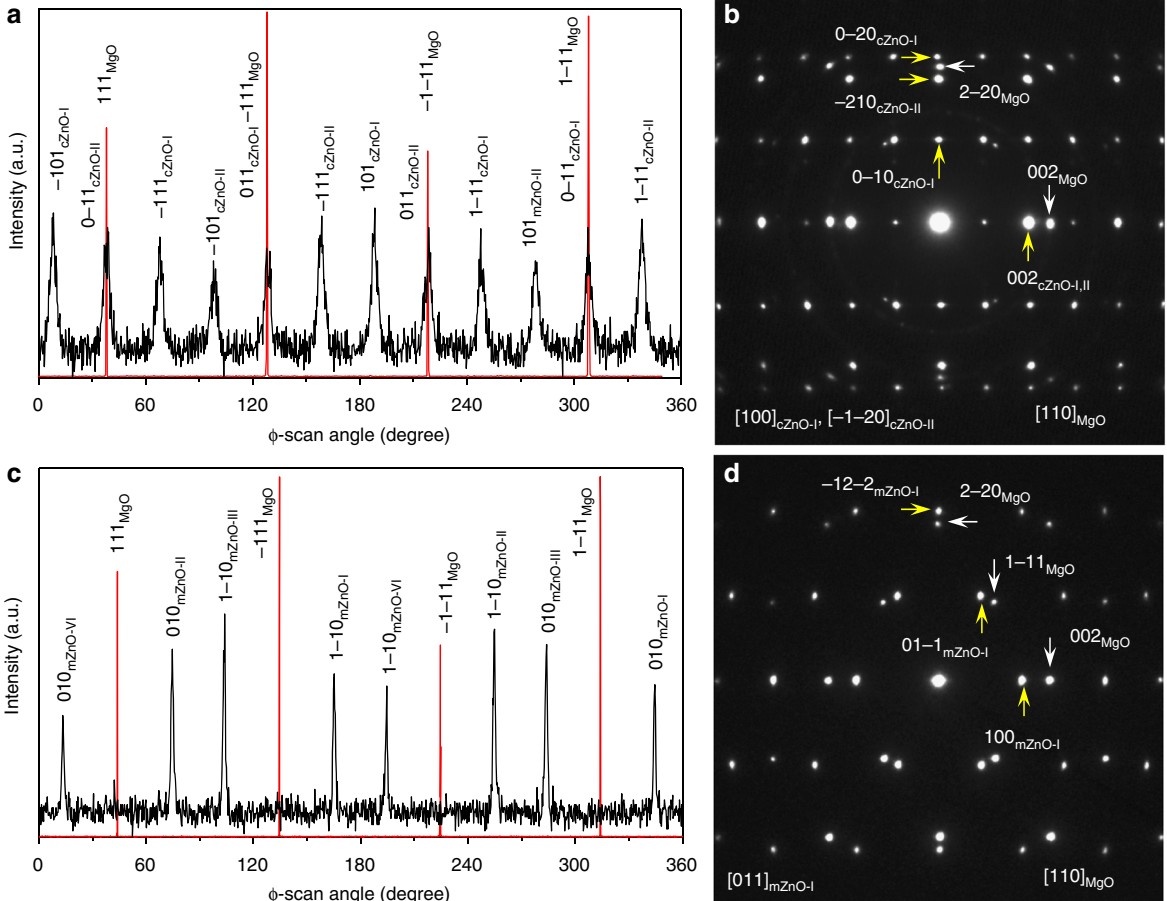

**Fig. 3** Orientation relationship for c-ZnO and m-ZnO. **a** X-ray diffraction (XRD) φ scan and (**b**) Electron diffraction pattern (EDP) from the c-ZnO film. The XRD φ-scan is carried out with $2\theta = 37°$, $\chi = 54.7°$ for MgO and $2\theta = 36.2°$, $\chi = 61.6°$ for c-ZnO. The EDP consists of three sets of patterns: $[110]_{MgO}$, $[100]_{cZnO\text{-}I}$ and $[−1−20]_{c\text{-}ZnO\text{-}II}$ (subscripts MgO, cZnO-I, and cZnO-II represent MgO substrate, c-ZnO domain I and II, respectively). **c** XRD φ-scan and (**d**) [011] zone EDP from the m-ZnO film. Subscripts MgO, mZnO-I, mZnO-II, mZnO-III, and mZnO-VI represent MgO, m-ZnO domains I, II, III, IV, respectively. The XRD φ-scan is carried out with $2\theta = 37°$, $\chi = 54.7°$ for MgO and $2\theta = 31.7°$, $\chi = 60°$ for m-ZnO

$010_{mZnO\text{-}II}$ and $1-10_{mZnO\text{-}II}$ with subscript mZnO-II representing m-ZnO domain II) deviate from the $111_{MgO}$ peak by 31° and 211°, respectively, indicating that $[010]_{mZnO\text{-}II}$ is parallel to $[140]_{MgO}$. Actually, the m-ZnO-II is rotated 90° along $[210]_{ZnO}$ direction (the normal of (001) plane) in terms of m-ZnO-I. In this case, the ZnO [08–3] is parallel to MgO [110]. Therefore, when the incident beam is along the MgO [110] direction, the m-ZnO-II would be viewed along a high index zone ($[08–3]_{mZnO\text{-}II}$) in which the projected distances among the atoms along $[011]_{mZnO\text{-}II}$ (vertical direction) are too close to be resolved. In this way, the streak-like pattern with a spacing of about 0.262 nm is formed. The calculated STEM-HAADF image along [08–3] direction (inset with white outline in Fig. 4b) agrees with the observation very well. The reconstructed interface structure based on XRD and STEM-HAADF is shown in the bottom part of Fig. 4c, d. The second orientation relationship is:

$$(100)_{mZnO\text{-}II}\left\|(001)_{MgO}, [010]_{mZnO\text{-}II}\right\|$$
$$[140]_{MgO}, [001]_{mZnO\text{-}II}\left\|[−410]_{Mgo}. \right. \tag{4}$$

From the third and the fourth pairs of 010 peaks in Fig. 3c, we conclude the third and the fourth orientation relationships for the

m-ZnO film:

$$(100)_{mZnO−III}\left\|(001)_{MgO}, [010]_{mZnO−III}\right\|$$
$$[1−40]_{MgO}, [001]_{mZnO−III}\left\|[410]_{MgO}. \right. \tag{5}$$

$$(100)_{mZnO−VI}\left\|(001)_{MgO}, [010]_{mZnO−VI}\right\|$$
$$[410]_{MgO}, [001]_{mZnO−VI}\left\|[−140]_{MgO}. \right. \tag{6}$$

Here, subscripts mZnO-III and mZnO-VI represent m-ZnO domains III and VI, respectively. The calculated STEM-HAADF images along with reconstructed interfacial structures are shown in Supplementary Fig. 3 with discussions in Supplementary Note 3. When the film is viewed along the MgO [110] direction for STEM-HAADF imaging, the m-ZnO domain III looks identical to the m-ZnO domain I, while the m-ZnO domain VI looks identical to m-ZnO domain II. These results can be further confirmed by the in-situ RHEED patterns (Supplementary Fig. 4 and Supplementary Note 4). Therefore, although XRD 0–360° φ-scan reveals four m-ZnO domains, STEM-HAADF imaging would only distinguish two of them.

The observation of four m-ZnO domains is consistent with 4 mm symmetry along [001] of the substrate MgO. Eight MgO <410> are equivalent, while ZnO [010] and [0–10] are

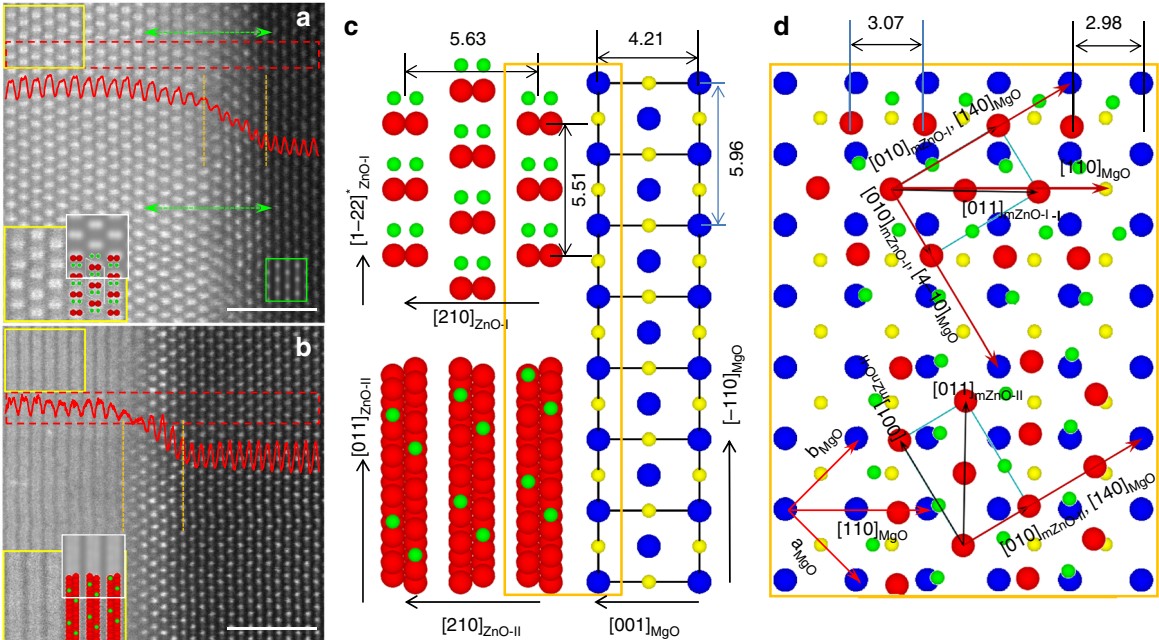

**Fig. 4** Orientation relationship for m-ZnO. Scanning transmission electron microscopy (STEM)- high-angle annual dark-field (HAADF) images of (**a**) m-ZnO-I and (**b**) m-ZnO-II domains of the m-plane film. Scale bar, 2 nm. The insets show the magnified images from the area outlined by the yellow rectangles with the atomic projection of $[011]_{mZnO-I}$ in **a** and $[0–83]_{mZnO-II}$ in **b** embedded. The inset outlined by white line squares in **a**, **b** are calculated STEM-HAADF images along $[011]_{ZnO}$ and $[08–3]_{ZnO}$, respectively. They agree with the experimental images very well. Note, there are 5 ~ 6 atomic layers in the interface that have an atomic arrangement similar to MgO, as marked by a pair of orange vertical dashed lines. The image intensity decreases from the ZnO film to the MgO substrate over these layers, as indicated by the embedded image intensity line scans (red) from the red dashed rectangles (integrated vertically). This indicates that there exists an interface phase $Zn_xMg_{1−x}O$ (x decreases from the ZnO film side to the MgO substrate side, see Supplementary Note 9 for electron energy loss spectroscopy (EELS) analysis). The two green dashed arrows in **a** indicate the alignment of a ZnO row with an MgO row. **c** Projection of the interface for m-ZnO-I along [011] direction (upper panel) and m-ZnO-II along [08–3] direction (lower panel) derived from STEM images. **d** Top view (one layer of a $(100)_{ZnO}$ plane on top of one layer of a $(001)_{MgO}$ plane) of the interface area outlined by the brown rectangle in **c** (rotated 90° clockwise). The m-ZnO-II domain (lower panel) can be obtained by rotating 90° from m-ZnO-I (upper panel)

equivalent. There would be equal probability for ZnO to align its [010] (or [0–10]) parallel to one of the MgO <410>, resulting in the formation of four independent m-ZnO domains. The occurrence of two rotational domains in c-ZnO/MgO and four rotational domains in m-ZnO/MgO, which is consistent with the theoretical prediction based on the mismatch of rotational symmetry at the interface (Supplementary Note 5). The observed interface relationship for m-ZnO shown in (Eqs. 3–6), as well as that of c-ZnO shown in (Eqs. 1 and 2), is confirmed by the first principles calculations to have the lowest interface energy among a range of the possible interface structures (see later discussions).

**Lattice coupling between the film and the substrate**. The d-spacing of MgO along <110> direction is equal to 2.98 Å, while those of ZnO along $[010]^*$ (or [120]) and $[2–10]^*$ (or [100]) are equal to 2.81 Å and 1.62 Å, respectively. The lattice mismatch is, therefore, calculated to be 5.7% along the $[120]_{ZnO}$ direction and 8.7% along the $[100]_{ZnO}$ direction in the case of c-ZnO. From Fig. 2a, the lattice coupling relationship between the film and the substrate at the interface for c-ZnO is: $17d_{010ZnO} = 16d_{110MgO}$ (or $8\frac{1}{2}[120]_{ZnO} = 8[110]_{MgO}$, see horizontal dashed lines in Fig. 2a). In addition, from the STEM-HAADF image (Fig. 2a) and EDP (Fig. 3b), the lattice parameters a and c were measured to be 3.24 ± 0.02 Å and 5.22 ± 0.02 Å, respectively, in the films. These are close to the values of bulk sample (a = 3.25 Å and c = 5.21 Å). That is to say, the lattice parameter $a_{ZnO}$ appears almost unchanged from the bulk to the film. For the c-ZnO film, the termination at the interface could be either oxygen-plane or zinc-plane. To determine the termination, we simultaneously acquired

STEM-HAADF and annular bright field (ABF) images for c-ZnO, as shown in Supplementary Fig. 5. The insets show the magnified images from the area outlined by the rectangles with the atomic projection of $[100]_{cZnO-I}$. While O contrast is weak and blur near the interface due to the strain, they are clearly seen in the right side of the Zn atoms in the magnified image, indicating that the ZnO film is along [001] direction, e.g., O is closer to the interface than Zn, which is consistent with our first principle calculations that the O-termination at the interface yields a lower interface energy (see discussions later). Supplementary Fig. 5 shows inverse FFT images by applying an aperture in $020_{ZnO}/−220_{MgO}$ spots of the FFT (inset) of STEM-HAADF image. Dislocations are clearly seen at the interface, as marked by T. From the simultaneously acquired STEM-HAADF and STEM-ABF images for c-ZnO film (Supplementary Fig. 5) with the interface on the right, sharper O contrast in the ABF image can be seen as the film in this area is free of defects. More detailed discussions can be found in Supplementary Note 6 and Supplementary Fig. 6, and Supplementary Note 7 shows that c-plane films are composed of two different kinds of 'nanorods' with a rotation of 30° between them (the corresponding boundary lines are marked by the red lines). These observations suggest the c-ZnO films are formed by the self-assembly growth model. The defects located at the interface between the ZnO layer and MgO layer as marked by the white arrows in Fig. 2a can be further confirmed by the examination of the enhanced contrast with color and the plot of peak intensities in 3D and profile, as shown in Supplementary Fig. 7 and described in Supplementary Note 8. Clemens et al.[27] demonstrated that there were many point defects on the MgO-(001) plane after the treatment of annealing owing to the Mg atoms

evaporation. But, it is beneficial for the nucleation with the 3-dimensional island model.

For the m-ZnO film, the lattice mismatch between the film and the substrate is about 7.4% and 5.5% along the $[-110]_{MgO}$ and $[110]_{MgO}$ directions, respectively. The lattice coupling relationship between the film and the substrate at the interface is: $13d_{1-22ZnO} = 6d_{110MgO}$ (see the atom rows bounded by the two green dashed arrows in Fig. 4a). From this and EDP in Fig. 3d, the d-spacing of ZnO along $[1-22]^*$ direction (in-plane) and $[100]^*$ (out-of-plane) were measured to be $1.375 \pm 0.02$ Å and $2.816 \pm 0.02$ Å, respectively, close to the corresponding bulk values (1.378 and 2.815 Å) too. The result that the lattice lengths are nearly equal to the bulk values can be further confirmed by the in-situ RHEED patterns (Supplementary Fig. 8 and Supplementary Note 9). Furthermore, from the STEM-HAADF image, the interface of c-ZnO is quite sharp, indicating an absence of diffuse areas of the substrate to ZnO film in the c-ZnO case. While in the case of m-ZnO, there is a transition area which has an atomic arrangement similar to MgO, but with stronger contrast, as marked by the orange dashed lines in Fig. 4 and Supplementary Fig. 6. This indicates there exists a transition phase (denoted as T-phase) of $Zn_xMg_{1-x}O$. This result had been confirmed by the EELS spectra (Supplementary Figs. 9 and 10, and Supplementary Notes 10 and 11). Apparently, the T-phase has a structure similar to cubic MgO, rather than ZnO. First principles calculations based on DFT results imply that the existence of T-phase on the MgO substrate will induce the decrements of the substrate surface energy, O diffusion barrier as well as the barrier to nucleation while increasing the Zn atom adsorption energy (see later discussions). To sum up, from the discussion about the STEM-HAADF images, the values of lattice parameter in films are close to the bulk value in spite of the existence of the large lattice mismatch and the interface for c-ZnO is sharper than that for m-ZnO.

**First principles calculations**. In this section, in order to investigate the mechanism of the transformation of the growth orientation and the stability of the interface structure between the ZnO film and the MgO (001) substrate, we carried out the first principle calculations based on DFT for the interface structures of two growth orientations. The corresponding calculation details can be found in the Supplementary Notes 12–16.

For polar plane of ZnO, it is "cleavage energy" instead of "surface energy" that can be defined as the cleavage of the bulk ZnO creates both (001)-Zn surface and (00–1)-O surface and no unique surface energy can be determined[28,29]. Here the "surface energy" is taken as half of the cleavage energy (using the values in Supplementary Table 2). The calculated "surface energy" values for c-, m- and a-plane of ZnO as well as the (001) MgO substrate surface are in good agreement with previously reported results also calculated by the DFT + GGA method[29–32]. For interface energy calculations, two interface models for c-ZnO with either O-termination or Zn-termination at the interface are considered and the calculations show a lower interface energy with O-termination than that with Zn-termination, in agreement with our experimental observation that the O-layer is closer to the MgO substrate (Supplementary Fig. 5). For the m-ZnO case, two interface models are also considered with and without the buffer layers between hexagonal ZnO and cubic MgO; for the model with buffer layer, the change of the substrate surface energy is considered as induced by the buffer layers, using a range between that of the (001)MgO and that of the (001)FCC ZnO, leading to a range of interface energy mentioned in Supplementary Note 14.

In order to understand why the experimentally observed interface registry follows the relationship as shown in (Eqs. 1 and 2) for c-ZnO/MgO and (Eqs. 3–6) for m-ZnO/MgO, we compared the total energies of a range of possible interface registry structures for the two growth orientations through theoretical calculations by the DFT method. Setting the experimentally observed registry structure as the origin point (orientation degree equals to 0°), a series of other possible registry structures can be obtained by the clockwise (anti-clockwise) rotation of the ZnO overlayers along the surface normal of the MgO substrate, defined for negative (positive) rotation degrees as shown in Supplementary Fig. 11. The blue and red curves plot the calculated related energies (using the 0° orientation as the zero energy reference) for c-ZnO and m-ZnO, respectively, for the possible interface registry structures from −15° to 15° rotations and demonstrate that the interfacial relationships observed in our experiments can yield the lowest interface energies. This confirms why we have obtained the robust interfacial relationships for ZnO/MgO interfaces.

Supplementary Fig. 12 show the adsorption energies for the Zn and O atoms (corresponding to the most stable saddle-point configurations shown in Supplementary Fig. 13), respectively, on the MgO (001) surface without and with the buffer layer. Remarkably, the adsorption energy for Zn atoms (higher than −0.1 eV) is far higher than that for O atoms (lower than −4.0 eV) on the MgO (001) surface without the buffer layer, indicating that it is rather easy for Zn atoms to desorb, and hence difficult to form an island with Zn atom as the termination layer of the ZnO film close to the interface. On the contrary, O atoms can adsorb solidly on the MgO substrate. When there exist a buffer layer on the MgO surface, the Zn atom adsorption energy becomes higher, as shown in Supplementary Fig. 12, indicating that the formation of the transition layer can hinder the Zn desorption. Whereas, the buffer layer has little impact on O adsorption energies. The diffusion barriers for Zn and O atoms (as shown in Supplementary Fig. 12) show that Zn atom diffusion barrier is rather small (lower than 0.025 eV) no matter whether there exists a transition phase or not on the MgO substrate. However, for O atoms, the diffusion barrier on the ideal MgO surface is much larger (about1.2 eV) and becomes smaller (about0.8 eV) after adding the buffer layer(s).

## Discussions

As experimentally demonstrated, the grown ZnO films on the MgO (001) substrates can exhibit either polar c- or non-polar m-directions depending on the growth conditions. The growth direction of the film was robust and exhibited a transition after nucleating, which can be demonstrated by in-situ annealing experiments (Supplementary Fig. 14 and Supplementary Note 17) under the higher temperature condition (around 673 K) with or without O₂. In fact, nucleation of the ZnO film on the MgO substrate from the gas phase is intrinsically a non-equilibrium phenomenon governed by a competition between kinetics and thermodynamics. Next, we discuss the mechanisms of the growth orientation alternatives from the viewpoints of thermodynamics and kinetics based on the first principles calculation results.

On the basis of the classical theory of nucleation[33–35], for nuclei in shape of clusters or islands that can be described in terms of macroscopic surface energy[36], the overall excess free energy $\Delta G$ for nucleation islands of ZnO on the MgO substrate can be obtained using the following equations

$$\Delta G = S_{interface} \cdot (\gamma_{interface} - \gamma_{substrate}) + S_{island} \cdot \gamma_{island} + V_{island} \cdot \Delta G_v \quad (7)$$

where $S_{interface}$ is the interface area of contact between the nucleation island and the substrate, $S_{island}$ and $V_{island}$ are the surface area and volume of nucleation island, respectively.

$\gamma_{\text{interface}}$ represents the interface energy per unit area, and $\gamma_{\text{substrate}}$ and $\gamma_{\text{island}}$, are the surface energy per unit area for substrate and island, respectively. $\Delta G_v$ is the free energy change of the island per unit volume. The values of interface energy ($\gamma_{\text{interface}}$), surface energy ($\gamma_{\text{substrate}}$) and free energy change ($\Delta G_v$) can be obtained from first principles calculations (Supplementary Tables 1–3), whereas, $\gamma_{\text{island}}$ depends on the shape of the island.

As shown by the atomic force microscopy (AFM) images in Supplementary Fig. 15), the island slope of the nucleation stage for c-ZnO (8°–14°) is much greater than for m-ZnO (1°–4°) after growing for 10 min. Therefore, the growth of the c-ZnO follows a three-dimensional-like mode, while that of m-ZnO is of a 2-dimensional-like mode. Additionally, the corresponding in-situ RHEED patterns, which show rather weak spots for the c-ZnO and yet bright stripes for the m-ZnO, available in the insets of Supplementary Fig. 15, show that the roughness of c-ZnO is far greater than that of m-ZnO, further confirming the difference of the contact angle of nucleation between c-ZnO and m-ZnO.

On the basis of AFM images, the morphology of c-ZnO and m-ZnO islands is estimated to be of typical polygonal shape, i.e., "frustum with facets" (with typical contact angles of 11° for c-ZnO and 3° for m-ZnO). However, it is not straightforward to obtain reliable shapes of small islands by AFM, due to convolution effects with the tip shape, therefore, we have also considered two additional possible shapes for the islands, namely "circular cone frustum" and "spherical cap" (Supplementary Figs. 16–18 and discussions in Supplementary Notes 18–21). Remarkably, all the three proposed morphological shapes lead to the same conclusion that, c-ZnO with a larger contact angle yields a smaller nucleation barrier than that of m-ZnO with a smaller contact angle. Therefore, the nucleation barrier seems to be more sensitive to the contact angle than the exact shape of the nuclei. The theoretical analysis is in good agreement with our experimental observation that c-ZnO films with a larger contact angle are grown at a lower temperature while m-ZnO films with a smaller contact angle exhibit at a higher temperature.

From the calculation results (Supplementary Figs. 11 and 12), we can find that during nucleating, the limit factor is adatom adsorption for Zn atoms and yet adatom diffusion for O atoms. Here, the buffer layers (the so-called T-phase) play a vital role for the adsorption energy and diffusion barrier. At a lower growth temperature, there appear no T-phase on the substrate and most of O adatoms adsorb solidly on the MgO surface with only little diffusion. Simultaneously, little Zn atoms can adsorb on the ideal MgO surface. Under this condition, there will appear nucleation with O-termination and with growth direction along [001] as well as a large contact angle. This interface feature with a rigid termination is similar to the self-assembled organic molecular nailing down the gold nanostructures[37]. On the contrary, if the growth temperature becomes higher, it will induce the atom diffusions and then form the T-phase on the MgO substrate, which hinders the desorption of Zn atoms and promotes the diffusion of O atoms. This result will stimulate the simultaneous presentation of Zn and O atoms on the termination layer of the ZnO film at the interface and thus decrease the contact angle between the nucleation and MgO substrate. Thereby the increment of growth temperature as well as the appearance of the T-phase will result in the occurrence of the growth orientation transformation. Additionally, the increment of $O_2$ partial pressure can inhibit the atom diffusion, and simultaneously rise the probability of the nucleation with O-termination. This can explain why even for the higher growth temperature (up to 600 K), the growth orientation of ZnO film still appears along the [001] azimuth with the increase of $O_2$ partial pressure (Fig. 1).

In summary, this work examines the mechanism of the growth orientation transformation. The transformation is tailored by the growth temperature and pressure simultaneously, resembling the phase transition of water to vapor-water. The interface structures of both c-ZnO/MgO and m-ZnO/MgO were thoroughly characterized by XRD, EDP, and STEM as well as first principles calculations based on DFT, revealing two rotational domains for the c-ZnO films and four rotational domains for the m-ZnO films and their interface registry stability. The mechanism of the growth orientation is examined from both thermodynamics and kinetics points of view. Thermodynamically, it is found that, the nucleation barrier for c-ZnO with a larger contact angle (e.g., $\geq 11°$) is lower than that of m-ZnO with a smaller contact angle (e.g., $\leq 4°$). Kinetically, it is found that, under the condition of lower growth temperature or larger $O_2$ pressure, with few Zn atoms adsorbing and few O atoms diffusing on the substrate, the nucleation pathway follows a three-dimensional model with a large contact angle and an O-termination at the interface, resulting in the growth along the c-plane direction. On the contrary, at a higher growth temperature, the Zn atom adsorption and O atom diffusion are both enhanced, which promotes the transformation of nucleation model from a 3-dimensional to a qausi-two-dimensional model, resulting in the growth along the m-plane direction. Both thermodynamic and kinetic mechanisms explain the diagram of growth direction transformation tailored by the growth temperature and pressure, as shown in Fig. 1. This work not only offers a clear image of the interfacial coupling between the cubic and wurtzite phase, but also proposes a reasonable theory model to explain the phenomena of the growth orientation transformations.

## Methods

The ZnO thin films were prepared on MgO (001) substrates by molecular beam epitaxy (MBE). Before being transferred into the MBE chamber for ZnO thin films growth, the MgO substrates were cleaned in an ultrasonic bath sequentially with acetone and ethanol for 5 min, respectively. Prior to the film growth, the MgO substrates were thermally cleaned at about 750 K for 60 min, with an oxygen pressure of $5 \times 10^{-5}$ mbar and a plasma power of 250 w. As shown in our previous report[7], the bright and streaky in-situ RHEED patterns from the treated MgO (001) surface indicated a smooth surface structure. In order to investigate the differences in the morphology between the two films with different growth directions, two series of ZnO films (namely polar and non-polar planes, respectively) were prepared. The polar films were grown at a substrate temperature of about 450 K and Zn (with a purity of 99.9999%) source temperature of 630 K with $1 \times 10^{-5}$ mbar of the oxygen pressure and 180 W of the plasma power. For the non-polar ZnO films the corresponding growth conditions are about 650 K (substrate temperature), 630 K (Zn source temperature), $1 \times 10^{-5}$ mbar (oxygen pressure) and 180 W (plasma power), respectively. The surface structures and the interface relationships between the films and substrates were analyzed by in-situ RHEED and ex-situ XRD techniques (using a Rigaku rotating anode X-ray generator and diffractometer). The interfacial atomic structure for the c-ZnO and m-ZnO films were examined by STEM-HAADF images, electron diffraction patterns and EELS spectra using double aberration corrected JEM-200CF microscope equipped with Gatan quantum energy filter and dual EELS. Image simulations were carried out using our own computer codes based on the multislice method with frozen phonon approximation. Finally, the growth mechanism of the ZnO thin films was studied through analyzing the evolution of the surface morphology by AFM.

**Calculation methods**. In this paper, we performed first principle calculations based on DFT[22] through VASP to examine the interface stability, including the calculations of surface energy, interface energy for two different growth orientations with a variety of interface structure for each orientation and compare with the observations from the STEM, RHEED and XRD results. The exchange and correlation effects were treated by the generalized gradient approximation (GGA)[23]. Projected augmented wave (PAW) potentials were employed with the cutoff of 500 eV for all the calculations. The EELS spectra were calculated using the TELNES package included in the WIEN2K code[38], a full potential linear augmented plane-wave plus local-orbitals method within DFT.

**Data availability**. The data that support the findings of this study are available from the corresponding authors upon request.

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

## Acknowledgements

This work is supported by the National Natural Science Foundation of China (Grant Nos. U1332105, 61227009, and 91321102), the Fundamental Research Funds for Central Universities (Grant No. 20720160020). The electron microscopy work at the Condensed Matter Physics and Materials Science Department, BNL was supported by the U.S. Department of Energy, Office of Basic Energy Science (BES), Materials Science and Engineering Division, and work at the Center for Functional Nanomaterials was supported by BES Scientific User Facilities Division, both were under Contract No. DE-SC0012704.

## Author contributions

H.Z. Carried out the growth and characterization experiments, performed the first principle calculations and drafted the manuscript. H-Q.W.: Led the project, analyzed the results, and revised the manuscript. L.W. and Y.Z.: Led the STEM/EELS experiments, data analysis and orientation relationship determination. J-C.Z.: Led the theoretical calculations and nucleation model analysis. L.W. and Y.L.: Analyzed the XRD data. Z.W., H.C., S.K., and Y.L.: Participated in the modeling and simulations. L.Z. and K.K.: Prepared the TEM samples and captured high resolution TEM images. J.K.: Provided guidelines and discussions for the MBE experiments.
