## [Peer Review File · Nature Communications]

Reviewers' Comments:

Reviewer #1 (Remarks to the Author)

Report on manuscript NCOMMS-16-20467

"Interfaces between hexagonal and cubic structures and their orientation alternatives"

by Hua Zhou et al.

The authors present a very thorough study of the growth of ZnO thin films on an MgO(001) substrate. They reveal the possibility of orienting ZnO film along either (10-10) (the so-called m-orientation) or along (0001) (the c-orientation) and, with a combination of several high resolution techniques (STEM-HAADF, electron diffraction, XRD ϕ -scan, EELS), they succeed in making a detailed description of the interface relationships. They accompany their experimental results by classical numerical calculations of surface and interface energies, which they enter into a model of classical nucleation theory to account for the differences in morphology and growth of the films in the two orientations.

I think that this study is conceptually very interesting because it deals with the conditions under which two materials of different symmetry (cubic for MgO and hexagonal for ZnO) can have a (semi)-coherent interface. The experimental part seems to be pushed to the state of the art limit, although it is difficult for me to fully judge it.

I appreciated the idea of rationalizing the experimental results with the concepts of crystal growth theory. However, the theoretical part of the paper, although relying on a good strategy, deserves important improvements. In its present state, it lacks technical details (so that no one will be able to reproduce the results), lacks explanations (so that most readers will have difficulties to follow the reasoning), and some arguments are questionable from a theoretical point of view.

- The parameters of the Buckingham potential used are not given. What are their values and where do they come from (literature? New parametrization?)? In the literature, MgO and ZnO parameters are certainly available. However, across the interface, which microscopic interaction is taken into account? Has there been an ab initio test on some simple models of ZnO-MgO interfaces proving that the parameters correctly account for interfacial interactions?

- Surface energies: the authors say that the values they find are in good agreement with DFT results (ref 30,33). At variance, I found that their ZnO(0001) surface energy is 25% higher than that of reference 33. Values found previously by other authors should be included in Table 1, so that the reader can make his/her own opinion on the agreement.

- The ZnO polar (0001) orientation: since this orientation is polar, the surface has to be compensated by removing a certain amount of ions, otherwise the surface energy diverges with the thickness of the system. Since there have been experimental and theoretical results showing that the surface morphology of the Zn or O terminations is rather complex (O. Dulub, U. Diebold, G. Kresse: PRL 90 (2003) 016102, Lauritsen et al. ACS Nano 5 (2011) 5987), the authors should clarify which hypothesis they make on the surface configuration when they calculate the surface energy. Moreover, using classical atomistic simulations obliges to simulate a stoichiometric (neutral) system. Hence, one can only determine the average of the Zn and O termination energies. When applied to the determination of the shape of ZnO particles, this raises a problem of knowing which atoms (Zn or O?) are at the interface (to calculate the interface energy), and which (O or Zn) are at the outer surface (to have the relevant surface energy). I guess that depending on the hypothesis made, results could be quite different and the presence of a mixed phase at the (0001) interface might also substantially change the interface energy with respect to that found with pure ZnO. Detailed comments should be made on this question.

- Interface energies: the authors have experimentally determined the epitaxial relationship at their interfaces, but what about the registry? Did they make a systematic theoretical determination of the registry which yields the lowest interface energy for all the considered interfaces?

- Classical nucleation theory: My main concern deals with ΔG_{inn} , defined as the inner energy per unit volume. A numerical value is given, without precision on how it was calculated. I may guess

that it is the difference of energy of a ZnO formula unit in the bulk and its constituent atoms (or molecules?) in the gas phase, normalized to the formula unit volume in the crystal. If it is so, while the LAMMPS set-up may give reliable result for the crystal, certainly it is not fitted to treat isolated neutral atoms, nor diatomic molecules (which have strongly reduced ionic charges). Where does the numerical value of ΔG_{inn} come from? Moreover, in the classical nucleation theory, the ΔG_{inn} term depends on what is called the supersaturation, which, in the present context, depends on the flux of atoms. Experimentally, one may tune the value of the supersaturation and this changes the kinetics of nucleation and growth. How does this enter the present analysis?

- Shape of the particles and contact angle: the authors find that ZnO particles are more 3-dimensional when they grow along (0001) and more 2-dimensional along (10-10) and they enter the angle value found experimentally into the nucleation model. In that sense, the claim "From this quantitative analysis, we can deduce that the model of nucleation for c-ZnO is generally a 3-dimensional island..." is misleading and should be removed. I think that the authors should make efforts to obtain a theoretical estimation of the contact angle, for example using the Wulff-Kaishev theory which gives equilibrium shapes of particles on a substrate, since interface, top and side facet energies have been calculated. It is true that growth shapes may in some cases differ from equilibrium shapes, but the comparison would still be informative and the estimation would better complement the experimental determination.

- Side energies: since the contact of the particles with the substrate is assumed to be circular (no faceting), what is the meaning of the "side" facets and how were their energies calculated? This question is linked to the understanding of the value of ΔG^* in equation (3).

- Temperature effects: I was unable to follow the reasoning, starting from line 256 to 291. It seems to me that there is confusion between three mechanisms which are all relevant with their own characteristics: nucleation, growth and diffusion. In particular for the latter, I cannot understand how one can say anything without calculating the barriers of diffusion. This whole part should be rewritten. More generally, the whole theoretical part should be structured into small paragraphs, to make it more easily readable and understandable.

- DFT simulation for EELS: (Supplementary material, section S7): the simulation of mixed oxides is a very complex matter, especially when the two oxide crystalline phases have not the same symmetry. Literature results show that, in the case of $\text{Zn}_x\text{Mg}_{1-x}\text{O}$, the structure remains cubic up to some 15% of Zn doping, becomes hexagonal for dopings larger than some 80% and in between no one knows. Moreover, even when the structure is known, the distribution of dopants on the cation sites is not known (full disorder? Phase separation? Short range order? Long range order?). Much more details are thus required to explain under which hypothesis the EELS spectra have been simulated and what justifies the hypothesis.

To conclude, my opinion is that the work is original, has potentialities to interest a large community, but that it is not yet ready to be published under the present form. I thus recommend major revisions (at least of the theoretical part).

Pr. Claudine NOGUERA

Reviewer #2 (Remarks to the Author)

In this paper, Zhou et al. report two interface structures between ZnO and MgO, by combining STEM, theoretical calculations, RHEED and AFM. From these detailed results, they further examined the growth mechanisms and orientation transformation for the ZnO-MgO system. This is an important topic in materials sciences, however, I think some of the data and discussions are unconvincing and controversy, which makes the paper very confusing. I can't accept this paper due to the following reasons.

First of all, many of the discussions and conclusions are inconsistent. While they fabricated m-ZnO with lower temperature described in the experimental part, and mentioned "...the nucleation barrier of the m-ZnO is always smaller than the c-ZnO..." in Line 235; They mentioned and concluded several times with opposite arguments, for example, that "...the nucleation of m-ZnO requires a higher growth temperature..." (Line 272), "... suggests that the barrier of nucleation for

c-ZnO is lower than that for m-ZnO..."(Line 255). And the phase diagram in the Figure 1 also shows nonpolar m-ZnO is stable under higher temperature range. Moreover, since the lattice mismatch of each orientation in the present study are so large that they should be categorized as incoherent interfaces, rather than semi coherent interfaces. In addition, since the authors mentioned "a transition phase of $Zn_xMg_{1-x}O$ " is formed for m-ZnO thin film. Then it does not make sense for the lateral discussions and explanations for the growth orientation transformation, by simply compare the two growth modes under the same conditions, because for m-ZnO, the substrate is not simply MgO anymore, but the transition phase. I think the authors should take such effect into account for the growth model. Besides, some questions and comments are listed below.

1. How did the authors get Figure 1? According to this diagram, both thin films fabricated in this study should be either c-ZnO or m-ZnO (the unit for growth temperature in the experimental part is misprinted)
2. For the polar c-ZnO, how did the authors determine the polarity? In the Figure 2, they draw a schematic of O-terminated interface, while in the simulation shown in Figure 5d, they used Zn-terminated interface model. They should clarify and discuss the polarity using ABF STEM technique.
3. The authors should also consider the case of (100)[001]ZnO//(001)[110]MgO interface. Although the calculations show that the interface energy is indeed higher, but it is still comparable with c-ZnO case, and has been obtained by Ref 11. Then why this growth did not occur? If the authors simply conclude it as "This discrepancy may stem from differences in the experimental conditions" (Line 140), then they must state clearly that their conclusions are only valid for the present experimental conditions.
4. The authors should show the data of in-situ annealing experiments if they want to conclude "the growth direction of the film was robust and exhibited a transition after nucleating" (Line 206) instead of stating data not shown in Line 207.
5. The quality of STEM images are poor. In Figure 2a, I can not see defect pointed by the arrow clearly. In Fig. S3b, I do not understand how they drew such a boundary line. In Fig. S8d and f, the schematic of contact angle would be misleading.

Reviewer #3 (Remarks to the Author)

This provides the most comprehensive and detailed study to date on hexagonal/cubic crystal structures, and in particular in the ZnO/MgO system. While other groups have demonstrated the control of polar and nonpolar orientations of ZnO for growth on cubic substrates such as MgO, this work presents the underlying theory to explain the behavior, which may be applied to other material systems (including perovskites, GaN, etc).

The work is highly detailed and of quality suitable for this journal, where a select group of researchers in this area will be very interested in the work. However, it is still not clear that the work will be of broader interest to the readership of the journal.

In particular, the question arises: "now that you have explained the underlying physics, is it useful?" Some further data and discussion would be helpful, including the following issues:

- 1) These material structures have a large lattice mismatch. While interface control to achieve a given crystalline orientation is achieved, there will likely be a high dislocation density that will impede application to an electronic/optical/other device. Could the authors comment on the dislocation density in these material structures? Given the detailed TEM, XRD, etc, measurements, this data should be available.
- 2) Since the article focus, and title, is on the interface between hexagonal and cubic structures, how does the reverse process work? Can the interface be controlled for growth of cubic materials on varying orientation of the hexagonal material? This will be critical in applying the model to applications that would require multilayer structures.

Authors' response to reviewers' comments (ID: NCOMMS-16-20467)

We thank the three reviewers for the valuable comments regarding our manuscript. We have studied the comments carefully and have carried out additional calculations and experiments accordingly to address the reviewers' concerns. In particular, we have included the new and extensive first principles calculations using the Vienna Ab Initio Simulation Package (VASP) based on density functional theory (DFT) to replace those from the empirical potential calculation using the Large-scale Atomic/Molecular Massively Parallel Simulator (LAMMPS) package in the previous version. The first principles calculations take account the atomic interactions at the interface (especially the interaction between ZnO and MgO across the interface) in a much better way than the empirical potential calculations; the new theoretical analysis covers both thermodynamics and kinetics points of view. The whole theoretical part has been rewritten as suggested by one of the reviewers. And the title has also been revised to be "Interfaces between hexagonal and cubic oxides and their structure alternatives". All the revisions in the manuscript have been marked in red. Five co-authors have been added in light of their contributions to the revised manuscript.

Our responses to the reviewers' comments, point-by-point, are listed below.

Reviewers' comments:

Reviewer #1 (Remarks to the Author):

Report on manuscript NCOMMS-16-20467

"Interfaces between hexagonal and cubic structures and their orientation alternatives"

by Hua Zhou et al.

The authors present a very thorough study of the growth of ZnO thin films on an MgO(001) substrate. They reveal the possibility of orienting ZnO film along either (10-10) (the so-called m-orientation) or along (0001) (the c-orientation) and, with a combination of several high resolution techniques (STEM-HAADF, electron diffraction, XRD ϕ -scan, EELS), they succeed in making a detailed description of the interface relationships. They accompany their experimental results by classical numerical calculations of surface and interface energies, which they enter into a model of classical nucleation theory to account for the differences in morphology and growth of the films in the two orientations.

I think that this study is conceptually very interesting because it deals with the conditions under which two materials of different symmetry (cubic for MgO and hexagonal for ZnO) can have a (semi)-coherent interface. The experimental part seems to be pushed to the state of the art limit, although it is difficult for me to fully judge it.

Authors' response:

We appreciate the reviewer's encouraging comments on the idea of our manuscript.

Comments from Reviewer #1 (cont.)

I appreciated the idea of rationalizing the experimental results with the concepts of crystal growth theory. However, the theoretical part of the paper, although relying on a good strategy,

deserves important improvements. In its present state, it lacks technical details (so that no one will be able to reproduce the results), lacks explanations (so that most readers will have difficulties to follow the reasoning), and some arguments are questionable from a theoretical point of view.

- The parameters of the Buckingham potential used are not given. What are their values and where do they come from (literature? New parametrization?)? In the literature, MgO and ZnO parameters are certainly available. However, across the interface, which microscopic interaction is taken into account? Has there been an ab initio test on some simple models of ZnO-MgO interfaces proving that the parameters correctly account for interfacial interactions?

Authors' response:

We thank the reviewer for pointing out the important improvement needed for the theoretical part. To address the concern, we have performed new and extensive density functional theory calculations to replace those based on empirical potentials to better treat the atomic interactions at the interface. We have also presented the related technical details in the Supplementary Information and included more discussions in the main text. In particular, for the growth mechanism for alternative orientations, we reshape the discussion part from both thermodynamics and kinetics points of view, through the combination of theoretical and experimental analysis.

Comments from Reviewer #1 (cont.)

- Surface energies: the authors say that the values they find are in good agreement with DFT results (ref 30,33). At variance, I found that their ZnO(0001) surface energy is 25% higher than that of reference 33. Values found previously by other authors should be included in Table 1, so that the reader can make his/her own opinion on the agreement.

Authors' response:

We thank the referee for the comments and suggestions. The references #30 and #33 in original manuscript are now Ref. 29 and Ref. 28 in the revised manuscript, respectively. The surface energy per area of c-plane ZnO is ranging from 1.7 to 2.15 J/m² (PBE in Ref. 29: 1.7 J/m², B3LYP in Ref. 28: 2.0 J/m², LDA in Ref. 29: 2.15 J/m²), depending on the exchange-correlation functional used. Our calculated cleavage energy per area of c-plane ZnO is 1.79 J/m², very close to 1.7 J/m² in Ref.29 using the same exchange-correlation functional (PBE). The surface energies of different surfaces of ZnO have been listed in Table 1 in the revised manuscript with values from literatures for comparison.

Comments from Reviewer #1 (cont.)

- The ZnO polar (0001) orientation: since this orientation is polar, the surface has to be compensated by removing a certain amount of ions, otherwise the surface energy diverges with the thickness of the system. Since there have been experimental and theoretical results showing that the surface morphology of the Zn or O terminations is rather complex (O. Dulub, U. Diebold, G. Kresse:PRL 90 (2003) 016102, Lauritsen et al. ACS Nano 5 (2011) 5987), the authors should

clarify which hypothesis they make on the surface configuration when they calculate the surface energy. Moreover, using classical atomistic simulations obliges to simulate a stoichiometric (neutral) system. Hence, one can only determine the average of the Zn and O termination energies. When applied to the determination of the shape of ZnO particles, this raises a problem of knowing which atoms (Zn or O?) are at the interface (to calculate the interface energy), and which (O or Zn) are at the outer surface (to have the relevant surface energy). I guess that depending on the hypothesis made, results could be quite different and the presence of a mixed phase at the (0001) interface might also substantially change the interface energy with respect to that found with pure ZnO. Detailed comments should be made on this question.

Authors' response:

We agree with the referee's comment on the ZnO polar (0001) orientation: *"since this orientation is polar, the surface has to be compensated by removing a certain amount of ions, otherwise the surface energy diverges with the thickness of the system"*. The problem of internal electric field of ZnO c-plane surface has been addressed according to the strategies and discussions by Meyer (Phys. Rev. B **69**, 045416 (2004)) and Wander *et al* (J. Chem. Phys. **115**, 2312-2316 (2001)), and the results of slab calculations have been extrapolated containing up to 20 layers. The calculated value of cleavage energy per area is close to what was reported in Ref.9, using another strategy, as shown in Table S15.2 in the revised Supplementary Information.

Based on our experimental results from reflection high energy electron diffraction (RHEED) and scanning transmission electron microscopy (STEM), it is found that there are no obvious appearances of reconstruction on the ZnO (0001) film surface and that the film is terminated with O atoms at the interface and Zn atoms at the outer surface (see the new ABF-STEM images in Fig. S5 of the revised Supplementary Information). In the theoretical calculations, we have considered both cases of O- and Zn- terminations at the interface and determined that the O-termination yields a lower interface energy than the Zn-termination, in agreement with the experimental observations. The corresponding details are presented in the revised main text and Supplementary Information. Our STEM results also indicate that there is no obvious mixed phase at the interface for c-ZnO, while for m-ZnO, the inter phase is more clearly to be seen (as shown in Fig. S6 of the revised Supplementary Information), which has been taken into account in the theoretical calculations.

Comments from Reviewer #1 (cont.)

- Interface energies: the authors have experimentally determined the epitaxial relationship at their interfaces, but what about the registry? Did they make a systematic theoretical determination of the registry which yields the lowest interface energy for all the considered interfaces?

Authors' response:

For the registry perpendicular to interface, namely, the registry related to the film termination

at the interface, as we aforementioned, it is found that for c-ZnO, the O atoms are at the interface, and Zn atoms are at the outer surface, observed from STEM results and confirmed by our first principles calculations.

For the azimuthal registry parallel to interface, we have investigated the possible configurations of the interface between the ZnO film and MgO substrate with different rotation angles. The experimentally observed interfacial structure is set as the origin point (azimuthal degree = 0°), then a series of other possible azimuthal registry structures are obtained by the clockwise (anti-clockwise) rotation of the ZnO overlayers along the surface normal of the MgO substrate, defined for negative (positive) azimuthal degrees in Figs. 5a and 5b. Through theoretical calculations, it is found that the experimentally observed azimuthal registry structure yields the lowest interface energy, for both the c-ZnO and m-ZnO cases, as shown in Supplementary Information.

Comments from Reviewer #1 (cont.)

- Classical nucleation theory: My main concern deals with ΔG_{inn} , defined as the inner energy per unit volume. A numerical value is given, without precision on how it was calculated. I may guess that it is the difference of energy of a ZnO formula unit in the bulk and its constituent atoms (or molecules?) in the gas phase, normalized to the formula unit volume in the crystal. If it is so, while the LAMMPS set-up may give reliable result for the crystal, certainly it is not fitted to treat isolated neutral atoms, nor diatomic molecules (which have strongly reduced ionic charges). Where does the numerical value of ΔG_{inn} come from? Moreover, in the classical nucleation theory, the ΔG_{inn} term depends on what is called the supersaturation, which, in the present context, depends on the flux of atoms. Experimentally, one may tune the value of the supersaturation and this changes the kinetics of nucleation and growth. How does this enter the present analysis?

Authors' response:

We agree with the reviewer's points. The numerical value of ΔG_{inn} (in our revised manuscript, we renamed it using the commonly used term, ΔG_v) is the energy difference between the ZnO bulk crystal and its constituent atoms (Zn and O) in the gas phase. As first principle calculations are used in our revised manuscript, it is convenient to obtain the energy for crystal and isolated atoms with the supercell approach.

We agree that, the ΔG_v term also depends on the flux of atoms, which has an effect on chemical reaction that undergoes changes in chemical potential. In the growth of ZnO films, it refers to either Zn-rich or O-rich cases. In our cases, the source temperature of Zn atoms is fixed for all the growth, and the range of oxygen partial pressures shown in Figure 1 usually lead to the growth of stoichiometric ZnO films for either c-ZnO and m-ZnO, based on our experience. Therefore, it shall not affect the conclusion from our present analysis.

Comments from Reviewer #1 (cont.)

- Shape of the particles and contact angle: the authors find that ZnO particles are more 3-dimensional when they grow along (0001) and more 2-dimensional along (10-10) and they

enter the angle value found experimentally into the nucleation model. In that sense, the claim “From this quantitative analysis, we can deduce that the model of nucleation for c-ZnO is generally a 3-dimensional island....” is misleading and should be removed.

I think that the authors should make efforts to obtain a theoretical estimation of the contact angle, for example using the Wulff-Kaishev theory which gives equilibrium shapes of particles on a substrate, since interface, top and side facet energies have been calculated. It is true that growth shapes may in some cases differ from equilibrium shapes, but the comparison would still be informative and the estimation would better complement the experimental determination.

Authors’ response:

We thank the reviewer for pointing out the misleading sentence, which we have removed from the revised manuscript.

We also appreciate the reviewer’s suggestion about the theoretical estimation of the contact angle. We have made such an effort using the Wulff-Kaishev theory (see Equation(2) in Claude R. Henry, Morphology of supported nanoparticles (review), Progress in Surface Science 80 (2005) 92–116):

$$\Delta h/h_i = E_{adh}/\gamma_i \quad (1)$$

Where, Δh , E_{adh} , h_i , and γ_i represent the truncated part at the interface, the adhesion energy, the central distance to the facet parallel to the interface and the corresponding surface energy, respectively. The contact angle can then be obtained through the Young–Dupré equation:

$$\cos\theta = E_{adh}/\gamma_i - 1 \quad (2)$$

The substitution of our DFT calculation results for E_{adh} , and γ_i leads to contact angles being larger than 90°, which conflicts with our experimental observations (from Atomic Force Microscopy images) that most of the slopes of ZnO island at the initial stage are less than 20°. Therefore, the Wulff-Kaishev theory is probably not applicable to our case.

Comments from Reviewer #1 (cont.)

- Side energies: since the contact of the particles with the substrate is assumed to be circular (no faceting), what is the meaning of the “side” facets and how were their energies calculated? This question is linked to the understanding of the value of ΔG^* in equation (3).

Authors’ response:

We thank the reviewer for raising the concern. The shape of nucleation in our model is assumed to be a type of parabolic quadratic surface. This model is closer to the experimental results compared with spherical model. In the manuscript, we divide the surface area into two parts: in-plane surfaces (perpendicular to the normal of the substrate) and out-plane surfaces (parallel to the normal of substrate, the so-called “side surface”). The surface energy of the nucleation is mainly attributed to the in-plane surface energy, which can be calculated by the DFT method (see S15); the side surface energy is estimated using the average value of the lowest index planes vertical to the top surface, which can also be calculated by the DFT method (see S15). In order to justify this approximation, we also calculate the surface energy based on the Gibbs-Wulff theorem and obtain the similar results. More details can be found in S13 and S14 in the revised Supplementary Information.

Comments from Reviewer #1 (cont.)

- Temperature effects: I was unable to follow the reasoning, starting from line 256 to 291. It seems to me that there is confusion between three mechanisms which are all relevant with their own characteristics: nucleation, growth and diffusion. In particular for the latter, I cannot understand how one can say anything without calculating the barriers of diffusion. This whole part should be rewritten. More generally, the whole theoretical part should be structured into small paragraphs, to make it more easily readable and understandable.

Authors' response:

We thank the reviewer for the comments and we have followed the reviewer's suggestion to re-structure the whole theoretical part into small paragraphs: first introduce our DFT calculations results for interface structure, followed by the discussion of growth mechanism from both thermodynamics and kinetics points of view. Both adsorption energy and diffusion energy barriers are calculated for the discussion of kinetics.

Comments from Reviewer #1 (cont.)

- DFT simulation for EELS: (Supplementary material, section S7): the simulation of mixed oxides is a very complex matter, especially when the two oxide crystalline phases have not the same symmetry. Literature results show that, in the case of $Zn_xMg_{1-x}O$, the structure remains cubic up to some 15% of Zn doping, becomes hexagonal for dopings larger than some 80% and in between no one knows. Moreover, even when the structure is known, the distribution of dopants on the cation sites is not known (full disorder? Phase separation? Short range order? Long range order?). Much more details are thus required to explain under which hypothesis the EELS spectra have been simulated and what justifies the hypothesis.

Authors' response:

We thank the reviewer for raising the concern. From the literatures we investigated the crystal symmetry of $Zn_xMg_{1-x}O$: for $x > 0.67$, the structure could be hexagonal; for $x < 0.34$, it could become cubic (Appl. Phys. Lett. 72, 2466 (1998); Phys. Stat. Sol. (a) 200, 361 (2003)); for $0.49 < x < 0.60$, the structure could be mixed (Phys. Stat. Sol. (a) 200, 361 (2003)). Even for pure ZnO or MgO, when grown on some special substrates with the cubic structure or at the interfaces, their structures could also become cubic or hexagonal (Appl. Phys. Lett. 84, 4562(2004); Appl. Phys. Lett. 82, 562(2003)). Hence it is possible that $Zn_xMg_{1-x}O$ at the interface appears cubic structure even for $x > 0.65$.

From our experiments, it is found that when the ZnO film is viewed along [011] (Fig. 4a) and [0-83] (Fig. 4b), the atomic projection of the interface layer of $Zn_xMg_{1-x}O$ is either the same or similar to that of the MgO [110], which indicates that the $Zn_xMg_{1-x}O$ inter phase is of cubic structure. Our DFT simulations for EELS are therefore based on the cubic structure for $Zn_xMg_{1-x}O$, where x is chosen to be 0.25 and 0.75 to show the transition from the MgO substrate to the ZnO film, as presented in Fig. S10 in the revised Supplementary Information.

Comments from Reviewer #1 (cont.)

To conclude, my opinion is that the work is original, has potentialities to interest a large community, but that it is not yet ready to be published under the present form. I thus recommend major revisions (at least of the theoretical part).

Pr. Claudine NOGUERA

Authors' response:

We thank the reviewer again for the insightful suggestions and comments, which are very hopeful for the improvement of our manuscript. We have rewritten the manuscript accordingly. And we hope that the major revision has addressed the reviewer's concerns.

Reviewer #2 (Remarks to the Author):

In this paper, Zhou et al. report two interface structures between ZnO and MgO, by combining STEM, theoretical calculations, RHEED and AFM. From these detailed results, they further examined the growth mechanisms and orientation transformation for the ZnO-MgO system. This is an important topic in materials sciences, however, I think some of the data and discussions are unconvincing and controversy, which makes the paper very confusing. I can't accept this paper due to the following reasons.

Authors' response:

We thank the reviewer for the valuable comments, which are helpful for us to further improve our manuscript.

Comments from Reviewer #2 (cont.):

First of all, many of the discussions and conclusions are inconsistent. While they fabricated m-ZnO with lower temperature described in the experimental part, and mentioned "...the nucleation barrier of the m-ZnO is always smaller than the c-ZnO..." in Line 235; They mentioned and concluded several times with opposite arguments, for example, that "...the nucleation of m-ZnO requires a higher growth temperature..."(Line 272), "... suggests that the barrier of nucleation for c-ZnO is lower than that for m-ZnO..."(Line 255). And the phase diagram in the Figure 1 also shows nonpolar m-ZnO is stable under higher temperature range.

Authors' response:

We thank the reviewer for reading carefully and for pointing out our inconsistent description and we apologize for the confusion caused. The m-ZnO films were fabricated at higher temperatures than the c-ZnO films, as shown in the phase diagram in Fig. 1. We have checked all the other descriptions and made corrections where necessary. About the comparison for nucleation barrier, we have clarified using the following sentence: "Additionally, owing to the c-ZnO having larger interface and surface energies compared with the m-ZnO, the nucleation barrier of the m-ZnO is always smaller than the c-ZnO for the same contact angle, as shown Fig. 7b. However, under the condition that the contact angle (e.g. $>11^\circ$) along the [001] azimuth is much larger than that along the $[210]_{\text{ZnO}}$ azimuth (e.g. $<4^\circ$), the nucleation barrier for c-ZnO will become smaller than that for m-ZnO, as indicated by the blue dashed lines in Fig. 7b, leading to a

preferred growth of the thin film along the [001] azimuth direction.”, in the revise manuscript.

Comments from Reviewer #2 (cont.):

Moreover, since the lattice mismatch of each orientation in the present study are so large that they should be categorized as incoherent interfaces, rather than semi coherent interfaces.

Authors' response:

We thank the reviewer for raising this concern. Generally speaking, the interface coupling between the film and the substrate often exhibits incoherent or semi-coherent characteristics instead of coherent when there is a large mismatch or difference of the symmetry between the film and the substrate. As far as we understand, the classification of interface does not only depend on the extent to which the lattice is mismatched, but also on how the lattice is actually connected at the interface. Figure R1 below (from Philosophical Magazine A, 75, 1329 (1997)) shows the schematic illustrations for the coherent, semi-coherent and incoherent interfaces. For a coherent interface, the lattice is matched perfectly at the interface plane, or the lattice mismatch is accommodated by the elastic strain (Fig. R1a); for a semi-coherent interface, misfit dislocations provide, at least, partial compensation of misfit stresses (Fig. R1b); while for an incoherent interface, it can be treated as a result from the rigid contact of two crystalline lattices (Fig. R1c). Our new STEM results show the existence of dislocations at the c-ZnO / MgO interface (see Fig. S5c in the Supplementary Information), as well as the continuous lattice in between dislocations, which is close to the semi-coherent interface feature at Fig. R1b. For the m-ZnO / MgO, the existence of a buffer layer (or inter phase) also indicates that the interface is probably not an incoherent case. On the other hand, we also noticed that some other references classify the interface type using interface energy (Ref: James Howe, Interface Materials (Wiley, America, 1997)), as shown in Table R1. Our calculated interface energy (Table 1 in the main text) falls in the range for incoherent interface, though. The fact that there appear rotational domains in both c-ZnO / MgO and m-ZnO / MgO makes the interface structure more complicated and the type of interface is not easy to define. To avoid confusions, we decided to remove the wording of “semi-coherent” in our manuscript. This shall not affect our main point of growth orientation transition.

Schematic drawing of (a) a coherent interface, (b) a semicoherent interface and (c) an incoherent interface.

Fig. R1

(Ref: Philosophical Magazine A, 75, 1329 (1997))

Table R1 Ranges of solid-solid interphase boundary energies for three types of planar interfaces.

Interface	γ^{SS} (mJ/m ²)
Coherent	5–200
Semicoherent	200–800
Incoherent	800–2500

(Ref: James Howe, Interface Materials (Wiley, America, 1997))

Comments from Reviewer #2 (cont.):

In addition, since the authors mentioned “a transition phase of $Zn_xMg_{1-x}O$ ” is formed for m-ZnO thin film. Then it does not make sense for the lateral discussions and explanations for the growth orientation transformation, by simply compare the two growth modes under the same conditions, because for m-ZnO, the substrate is not simply MgO anymore, but the transition phase. I think the authors should take such effect into account for the growth model.

Authors' response:

We thank the reviewer for the insightful comments. In our new DFT calculations, we have taken into account the transition phase for m-ZnO during the calculations of interface energies, adsorption energies and diffusion barriers, as well as for the discussions of growth models. All can be found in the new theoretical section of the revised manuscript.

Comments from Reviewer #2 (cont.):

Besides, some questions and comments are listed below.

1. How did the authors get Figure 1? According to this diagram, both thin films fabricated in this study should be either c-ZnO or m-ZnO (the unit for growth temperature in the experimental part is misprinted)

Authors' response:

We thank the reviewer for raising the good question. The phase diagram in Figure 1 was generated based on a series of XRD data and RHEED patterns, which we have added in S1 in the revised Supplementary Information.

Comments from Reviewer #2 (cont.):

2. For the polar c-ZnO, how did the authors determine the polarity? In the Figure 2, they draw a schematic of O-terminated interface, while in the simulation shown in Figure 5d, they used Zn-terminated interface model. They should clarify and discuss the polarity using ABF STEM technique.

Authors' response:

We have followed the reviewer's valuable suggestion and performed ABF STEM to determine the polarity of c-ZnO, as shown in the new session of S5 in the revised Supplementary Information. It is found that, the c-ZnO film is terminated by the O-plane at the interface. This is also confirmed by our new DFT calculations, which shows a lower interface energy for O-termination interface than that for Zn-termination interface, as shown in the added Table 2 of the revised manuscript.

Comments from Reviewer #2 (cont.):

3. The authors should also consider the case of $(100)[001]ZnO//[(001)[110]MgO$ interface. Although the calculations show that the interface energy is indeed higher, but it is still comparable with c-ZnO case, and has been obtained by Ref 11. Then why this growth did not occur? If the authors simply conclude it as “This discrepancy may stem from differences in the

experimental conditions" (Line 140), then they must state clearly that their conclusions are only valid for the present experimental conditions.

Authors' response:

We thank the reviewer for sharing the thoughts. In the revised manuscript and Supplementary Information, we presented our new theoretical calculations based on DFT. All the possibilities for the interface structures of c-ZnO and m-ZnO are considered and the new calculation results show the interface energy for the case of $(100)[001]\text{ZnO} // (001)[110]\text{MgO}$ in m-plane ZnO is larger than that of $(100)[001]\text{mZnO} // (001)[140]\text{MgO}$ and thus is not observed in our m-ZnO film. (For the same growth direction, the interface structure shall prefers the case with the lowest interface energy.)

We have carefully re-visited the analysis in the original Ref. 11 (Cagin, E. *et al. Appl. Phys. Lett.* 92, 233505 (2008).) and would like to share our viewpoints regarding the interface relationship, different from that proposed by the authors. The pole figure in their paper (see Fig. R2a below) shows eight distinct reflections, which is actually similar to what we observed in our XRD phi-scan for mZnO. And their proposed relationship based on the pole figure is hard to justify because the MgO reflections are not shown. For their SADP pattern, they mentioned: "The SADP does not offer a clear indication of the in-plane lattice relationships", however, a careful inspection on their SADP by enhancing the contrast (Fig. R2b below) reveals weak spots as indicated by the red arrow and circles. Using the spots of MgO as reference (note, the spot indexed as -200MgO should be indexed as 001MgO based on m-plane $(100)\text{ZnO} // (001)\text{MgO}$), the weak spots could be indexed as ZnO $[012]$ (or $[-12-33]$ in Bravais-Miller index) zone pattern. Therefore, the ZnO $[012]$ should roughly be parallel to MgO $[100]$, which is inconsistent with their proposed $(100)[001]\text{ZnO} // (001)[110]\text{MgO}$ relationship because the angle between ZnO $[012]$ and $[001]$ is about 17° , while that between MgO $[100]$ and $[110]$ is 45° . Interestingly, this SADP can be well explained by our determined interface relationship, e.g. $(100)[001]\text{mZnO-III} // (001)[410]\text{MgO}$, as shown in Fig. R2(c). With the mZnO-III (m3) relationship, the angle between ZnO $[012]$ (thick red arrow) and MgO $[100]$ (thick black arrow) is about 3° . When the beam is aligned with MgO $[100]$ zone, ZnO is about 3° off the $[012]$ zone. Weak ZnO $[012]$ zone spots could be observed in this case. To avoid the confusion, we have chosen to delete the discussions related to Ref. 11.

Fig. R2 (a,b) Figures from Cagin, E. *et al. Appl. Phys. Lett.* 92, 233505 (2008); contracts in the SADP pattern has been enhanced for closer investigation. (c) Top view of mZnO-III domain

from Fig. S3 of our manuscript (Supplementary Information). The thick black and red arrows point to the MgO [100] and mZnO-III [012] directions, respectively.

Comments from Reviewer #2 (cont.)

4. The authors should show the data of in-situ annealing experiments if they want to conclude “the growth direction of the film was robust and exhibited a transition after nucleating” (Line 206) instead of stating data not shown in Line 207.

Authors’ response:

We thank the reviewer for the insightful comments and we have followed the suggestion to include the data of in-situ annealing experiments in the new session of S11 in the revised Supplementary Information.

Comments from Reviewer #2 (cont.):

5. The quality of STEM images are poor. In Figure 2a, I can not see defect pointed by the arrow clearly. In Fig. S3b, I do not understand how they drew such a boundary line. In Fig. S8d and f, the schematic of contact angle would be misleading.

Authors’ response:

We have addressed the reviewer’s concerns as follows:

- (1) For Fig. 2a, the image contrast acquired by high angle annular dark field detector in STEM (HAADF-STEM) is approximately proportional to $Z^{1.7}$ (Z: atomic number). Because the Mg (Z=12) atom is much lighter than Zn (Z=30), it appears rather dark and difficult to be seen. In light of the reviewer’s comments, we show the same image of Fig. 2a with false color, 3D perspective view and peak intensity profile in the new Fig. S7 in the revised Supporting Information, for a better vision of the indicated point defects.
- (2) For Fig. S3b (renamed as Fig. S6b in the revised Supporting Information), we have redrawn two lines to show a boundary zone between the two different domains in c-ZnO.
- (3) For Fig. S8d and f (renamed as Fig. S12d and f in the revised Supporting Information), the profiles were generated by atomic force microscopy image processing software to show the vertical feature. The height and thus the contact angles are for illustration only and does not represent the actual sizes. We have added the note in the figure caption to avoid confusion.

Reviewer #3 (Remarks to the Author):

This provides the most comprehensive and detailed study to date on hexagonal/cubic crystal structures, and in particular in the ZnO/MgO system. While other groups have demonstrated the control of polar and nonpolar orientations of ZnO for growth on cubic substrates such as MgO, this work presents the underlying theory to explain the behavior, which may be applied to other material systems (including perovskites, GaN, etc).

Authors’ response:

We appreciate the reviewer’s encouraging comments.

Comments from Reviewer #3 (cont.):

The work is highly detailed and of quality suitable for this journal, where a select group of researchers in this area will be very interested in the work. However, it is still not clear that the work will be of broader interest to the readership of the journal.

In particular, the question arises: "now that you have explained the underlying physics, is it useful?"

Authors' response:

We thank the reviewer for the comments. For the fabrication of heterojunction, ideally, a small mismatch or the same symmetry between the film and the substrate would be favored. However, there has raised the demand of the integration between two materials that have distinct structure symmetry. For example, the coupling between the functional and optoelectronic materials have attracted much attentions due to the characteristics of the electronic conduction properties tuned by the photonic situation (reference: ACS. nano: 6, 6242(2014), *Adv. Funct. Mater.* **2011**, 21, 2423–2429)). Many optoelectronic materials are of hexagonal structure, while the typical functional perovskite materials (e.g. SrTiO₃, BaTiO₃, etc), are of cubic structure. Thus, our work aims to provide some guidelines for the integration between these two groups of materials and could also provide some hints to the more general case of the interface with a larger mismatch or different symmetry. In fact, this is an important issue that has simulated pioneer research (e.g. Grundmann *et al*, Phys. Rev. Lett. 105, 146102).

Comments from Reviewer #3 (cont.):

Some further data and discussion would be helpful, including the following issues:

1) These material structures have a large lattice mismatch. While interface control to achieve a given crystalline orientation is achieved, there will likely be a high dislocation density that will impede application to an electronic/optical/other device. Could the authors comment on the dislocation density in these material structures? Given the detailed TEM, XRD, etc, measurements, this data should be available.

Authors' response:

We thank the reviewer for the comments and suggestions. To estimate the dislocations, we added Fig. S5c, which is a fringe image obtained by applying an aperture in 020_{ZnO}/₋₂₂₀_{MgO} spots of the FFT of STEM-HAADF image. Dislocations are clearly seen, as indicated by the 'T'. Through the careful investigation of the corresponding dislocation network in Fig. S5d, an average dislocation density of 1 dislocation per 8[-110]MgO, or 1 dislocation per 32 MgO (-220) lattice fringes can be estimated, as discussed in the revised Supplementary Information.

Comments from Reviewer #3 (cont.):

2) Since the article focus, and title, is on the interface between hexagonal and cubic structures, how does the reverse process work? Can the interface be controlled for growth of cubic materials on varying orientation of the hexagonal material? This will be critical in applying the model to applications that would require multilayer structures.

Authors' response:

We thank the reviewer for raising the point. Our research group has indeed successfully grown cubic films (e.g. SrTiO₃, NiO) on the hexagonal ZnO substrate (see Fig. 3R below for the NiO(001)/ZnO(0001)) and have investigated the occurrence of the rotational domain. As pointed out in the aforementioned reference (Grundmann *et al*, Phys. Rev. Lett. 105, 146102), "In the case of a mismatch of rotational symmetry, the number of rotation domains of material A on material B is different from that of B on A. A larger number of rotation domains can occur due to domain structure or nearly fulfilled additional symmetries of the substrate surface." We will continue the exploration of the growth of cubic films on the different orientations of the hexagonal substrates for our future research.

Fig. 3R (Yaping Li, Hui-Qiong Wang* et al; unpublished)

Report on revised manuscript NCOMMS-16-20467
"Interfaces between hexagonal and cubic oxides and their structure alternatives"
by Hua Zhou et al.

The authors have substantially modified their manuscript in response to my concerns. In various places, they have given the additional information which was needed and they have performed ab initio calculations of surface, interface and adhesion energies, to replace their previous estimates by classical atomistic methods. They also have calculated diffusion barriers of adatoms to support the observed temperature effects. The theoretical part is thus strengthened and easier to read.

I still have concerns about the hypothesis of a “parabolic quadratic” shape of the nuclei in the application of the classical nucleation theory. It seems to me very strange when applied to crystalline particles and I am not convinced that the AFM images provided in Fig.S12 really support such hypothesis. Indeed, it is well-known that it is extremely difficult to obtain reliable shapes of small particles by AFM, due to convolution effects with the tip shape. Considering a polygonal shape (with planar lateral facets) obeying the Wulff-Kaishev theory would have been much more physical, simpler to apply and would have led to less controversial results. The useless hypothesis of a “parabolic quadratic” shape leads to a succession of approximations in the two derivations of the nucleation barrier (S13 and S14) which are difficult to follow and accept. Moreover, in S14, the introduction of an (undefined and likely meaningless) σ_i obeying the Wulff relationship is very questionable, since Wulff theorem only applies to the *global* shape of a particle.

A second concern is linked to the simulations of interface azimuthal registry. The authors should give precisions on the size of the coincidence unit cell that they have used. It seems to me that, for small rotational angles, this size has to be pretty large. Is it really tractable ab initio? Have the authors made additional approximations which are not described in the text to tackle it?

Aside from these fundamental points, some parts of the manuscript have been written (or re-written) without enough care. Some sentences tell exactly the opposite of what one would think and there are inconsistencies at some places. For examples:

- In the abstract: “Nevertheless, the detailed mechanism of the transformation has yet been fully explored” likely means “Nevertheless, the detailed mechanism of the transformation has *not* yet been fully explored”
- In Figure 6c and d, in the insets: “without 1 layer buffer” and “without 2 layer buffer” should likely be replaced by “*with* 1 layer buffer” and “*with* 2 layer buffer”
- In Figure S15.2, the representations of the ZnO c-plane with O termination and that of the Zn termination are incorrect. The surface layer should contain only half of the O or Zn atoms, otherwise polarity is not compensated.
- In the caption of Figure S15-4, and line 485, saddle point configurations of Zn are found in (a-i) and those for O in (i-l) (instead of “saddle-point configurations for Zn and O adatoms (a-c) and (j-l) Zn and O adatoms”, which has no meaning.
- In the same Figure, one can read that oxygen adsorption energy is equal to 2.253eV in configuration j, while, in the main text, it is said (line 228) that adsorption energies of oxygen atoms are larger than 4 eV

I would like to say that this list is likely not exhaustive and the authors should check their manuscript more carefully. And they should correct their English or make it correct by

someone with English or American mother tongue. Even many technical terms are incorrect (like “nucleation” used instead of “nucleus”, “Regoin” instead of “Region” in Figure 1, “volume energy per area” instead of cohesion energy per unit volume”, etc)

To summarize, I think that, despite obvious improvements, the revised manuscript, both in its present form and content, is not yet ready for being accepted.

Pr. Claudine NOGUERA

Reviewer #1:

Remarks to the Author:

Comments attached

Reviewer #3:

Remarks to the Author:

The authors have provided a detailed response to my concerns described in the original review, and I believe that the article is ready to publish in present form.

Reviewer #4:

Remarks to the Author:

In this manuscript, authors give the comprehensive characterization of the ZnO/MgO interface structures by STEM and XRD. I am not familiar with the theoretical calculation, so my comments will be mainly focused on the experimental part.

First I want to say that the quality of the STEM work is impressive. They belong to the best results we can get using present microscopes.

Although authors modified the title, but I still think the title is too broad. why not specify the ZnO and MgO in the title?

Authors indexed the electron diffraction patterns in Fig. 3d as [011]ZnO and [110]MgO. The index for ZnO may be wrong. Based on the pattern, the incident electron beam direction should be along [1 -2 1 -3] direction in four-index, while [011] corresponding to [0 1 -1 1].

There are a lot of domains in both C-ZnO and m-ZnO, how large they are? Does the domain size relate to the deposition temperature? or interface strain?

Reviewer #5:

Remarks to the Author:

I have reviewed the corrections to the paper and the authors' reply to the comments and I can recommend publication of the manuscript as is.

Reviewer #3:

Remarks to the Author:

The authors have substantially modified their manuscript in response to my concerns. In various places, they have given the additional information which was needed and they have performed ab initio calculations of surface, interface and adhesion energies, to replace their previous estimates by classical atomistic methods. They also have calculated diffusion barriers of adatoms to support the observed temperature effects. The theoretical part is thus strengthened and easier to read.

I still have concerns about the hypothesis of a “parabolic quadratic” shape of the nuclei in the application of the classical nucleation theory. It seems to me very strange when applied to crystalline particles and I am not convinced that the AFM images provided in Fig.S12 really support such hypothesis. Indeed, it is well-known that it is extremely difficult to obtain reliable shapes of small particles by AFM, due to convolution effects with the tip shape. Considering a polygonal shape (with planar lateral facets) obeying the Wulff-Kaishev theory would have been much more physical, simpler to apply and would have led to less controversial results. The useless hypothesis of a “parabolic quadratic” shape leads to a succession of approximations in the two derivations of the nucleation barrier (S13 and S14) which are difficult to follow and accept. Moreover, in S14, the introduction of an (undefined and likely meaningless) σ ; obeying the Wulff relationship is very questionable, since Wulff theorem only applies to the *global* shape of a particle.

A second concern is linked to the simulations of interface azimuthal registry. The authors should give precisions on the size of the coincidence unit cell that they have used. It seems to me that, for small rotational angles, this size has to be pretty large. Is it really tractable ab initio? Have the authors made additional approximations which are not described in the text to tackle it?

Aside from these fundamental points, some parts of the manuscript have been written (or rewritten) without enough care. Some sentences tell exactly the opposite of what one would think and there are inconsistencies at some places. For examples:

- In the abstract: “Nevertheless, the detailed mechanism of the transformation has yet been fully explored” likely means “Nevertheless, the detailed mechanism of the transformation has *not* yet been fully explored”
- In Figure 6c and d, in the insets: “without 1 layer buffer” and “without 2 layer buffer” should likely be replaced by “*with* 1 layer buffer” and “*with* 2 layer buffer”
- In Figure S15.2, the representations of the ZnO c-plane with O termination and that of the Zn termination are incorrect. The surface layer should contain only half of the O or Zn atoms, otherwise polarity is not compensated.
- In the caption of Figure S15-4, and line 485, saddle point configurations of Zn are found in (a-i) and those for O in (i-l) (instead of “saddle-point configurations for Zn and O adatoms (a-c) and (j-l) Zn and O adatoms”, which has no meaning.
- In the same Figure, one can read that oxygen adsorption energy is equal to 2.253eV in configuration j, while, in the main text, it is said (line 228) that adsorption energies of oxygen atoms are larger than 4 eV

I would like to say that this list is likely not exhaustive and the authors should check their manuscript more carefully. And they should correct their English or make it correct by

someone with English or American mother tongue. Even many technical terms are incorrect (like “nucleation” used instead of “nucleus”, “Regoin” instead of “Region” in Figure 1, “volume energy per area” instead of cohesion energy per unit volume”, etc)

To summarize, I think that, despite obvious improvements, the revised manuscript, both in its present form and content, is not yet ready for being accepted.

Pr. Claudine NOGUERA

Reviewer #3:

Remarks to the Author:

The authors have provided a detailed response to my concerns described in the original review, and I believe that the article is ready to publish in present form.

Reviewer #4:

Remarks to the Author:

In this manuscript, authors give the comprehensive characterization of the ZnO/MgO interface structures by STEM and XRD. I am not family with the theoretical calculation, so my comments will be mainly focused on the experimental part.

First I want to say that the quality of the STEM work is impressive. They belong to the best results we can get using present microscopes.

Although authors modified the title, but I still think the title is too broad. why not specify the ZnO and MgO in the title?

Authors indexed the electron diffraction patterns in Fig. 3d as [011]ZnO and [110]MgO. The index for ZnO may be wrong. Based on the pattern, the incident electron beam direction should be along [1 -2 1 -3] direction in four-index, while [011] corresponding to [0 1 -1 1].

There are a lot of domains in both C-ZnO and m-ZnO, how large they are? Does the domain size relate to the deposition temperature? or interface strain?

Reviewer #5:

Remarks to the Author:

I have reviewed the corrections to the paper and the authors' reply to the comments and I can recommend publication of the manuscript as is.

Authors' response to reviewers' comments (ID: NCOMMS-16-20467A)

We thank the valuable comments from the four reviewers for the 2nd round of review regarding our manuscript entitled "Interfaces between hexagonal and cubic oxides and their structure alternatives" (ID: NCOMMS-16-20467A). We have studied the comments carefully and have carried out additional theoretical analysis and additional first principles calculations accordingly to address the reviewers' concerns.

Our responses to the reviewers' comments, point-by-point, are listed below.

Reviewers' comments:

Reviewer #1 (Remarks to the Author):

The authors have substantially modified their manuscript in response to my concerns. In various places, they have given the additional information which was needed and they have performed ab initio calculations of surface, interface and adhesion energies, to replace their previous estimates by classical atomistic methods. They also have calculated diffusion barriers of adatoms to support the observed temperature effects. The theoretical part is thus strengthened and easier to read.

Authors' response:

We appreciate the reviewer's encouraging comments on our substantial modification to our manuscript involving the new ab initio calculations.

I still have concerns about the hypothesis of a "parabolic quadratic" shape of the nuclei in the application of the classical nucleation theory. It seems to me very strange when applied to crystalline particles and I am not convinced that the AFM images provided in Fig.S12 really support such hypothesis. Indeed, it is well-known that it is extremely difficult to obtain reliable shapes of small particles by AFM, due to convolution effects with the tip shape. Considering a polygonal shape (with planar lateral facets) obeying the Wulff-Kaishev theory would have been much more physical, simpler to apply and would have led to less controversial results. The useless hypothesis of a "parabolic quadratic" shape leads to a succession of approximations in the two derivations of the nucleation barrier (S13 and S14) which are difficult to follow and accept. Moreover, in S14, the introduction of an (undefined and likely meaningless) σ obeying the Wulff relationship is very questionable, since Wulff theorem only applies to the global shape of a particle.

Authors' response:

We thank the reviewer for raising the concern. For the shape of the nuclei, we have followed the reviewer's suggestion to consider polygonal shape (named as "frustum with facets" in our revised manuscript and supplementary information) to replace our previous hypothesis of "parabolic quadratic" shape. In addition, we have also added two other possible shapes for nuclei, namely "circular cone frustum" and "spherical cap". All these models lead to the same conclusion as in our last version of manuscript, *i.e.* c-ZnO with a larger contact angle yields a smaller

nucleation barrier than that of m-ZnO with a smaller contact angle. Therefore, the nucleation barrier seems to be more sensitive to contact angle than the exact shape of the nuclei.

A second concern is linked to the simulations of interface azimuthal registry. The authors should give precisions on the size of the coincidence unit cell that they have used. It seems to me that, for small rotational angles, this size has to be pretty large. Is it really tractable ab initio? Have the authors made additional approximations which are not described in the text to tackle it?

Authors' response:

We agree with the reviewer's comment that, for small rotational angles, the size of the coincidence unit cell would increase significantly. Therefore we did not use superlattice modeling for the interface azimuthal registry. Instead, we used cluster models that would extend the flexibility of rotations.

In the last version of manuscript, we used "pyramid" models (see Fig. R1 (a)-(b) for c-ZnO and Fig. R2 (a)-(b) for m-ZnO, respectively). The experimentally observed registry structure is set as the origin point (orientation degree = 0°). The cluster for c-ZnO consists of 68 atoms (39 O plus 29 Zn atoms) and that for m-ZnO consists of 66 atoms (30 O plus 36 Zn atoms). For both cases, a 4-layer thick slab size of the MgO substrate is used with a 4×4×2 cell consisting of 256 atoms along with a vacuum region of 25Å. The calculation results for c-ZnO and m-ZnO are shown in Fig. R3a and Fig. R3c, respectively.

In the revised manuscript, we used more complex "hydrogen saturation" models (see Fig. R1 (c)-(d) for c-ZnO and Fig. R2 (c)-(d) for m-ZnO, respectively). The c-ZnO cluster is chosen to consist of 115 atoms (39H+ 50O +26 Zn), while that of m-ZnO consists of 142 atoms (45H+ 62O +35 Zn). The slab size of the MgO substrate as well as the vacuum region remain the same. The calculation results for c-ZnO and m-ZnO are shown in Fig. R3b and Fig. R3d, respectively.

It can be seen that, the two different cluster models yield the same trend, *i.e.* the experimentally observed registry structure has the lowest energy. We have also performed additional calculations using superlattice models for c-ZnO (for 0 degree and 15 degree as shown in Fig. R4 (a) and (b), corresponding to the interface relationship of $[-110][110](001)\text{MgO} // [100][-1-20](001)\text{ZnO}$ and $[100][010](001)\text{MgO} // [100][-1-20](001)\text{ZnO}$, respectively) and for m-ZnO (for +5 degree, -14 degree, +31 degree, as shown in Fig. R4 (c)-(e), corresponding to the interface relationship of $[1-30][310](001)\text{MgO} // [010][001](100)\text{ZnO}$, $[100][010](001)\text{MgO} // [010][001](100)\text{ZnO}$ and $[1-10][110](001)\text{MgO} // [010][001](100)\text{ZnO}$, respectively). It is also confirmed that the 0 degree case for c-ZnO and the +5 degree case for m-ZnO have the lowest energy among the superlattice models.

We would like to emphasize that, the point of our theoretical analysis is to figure out the "relative values" for the purpose of comparison instead of the accurate "absolute values". In light of this, several assumptions have been made in order to tackle the difficulties of the theoretical modeling and calculations for the complex interface systems. And we have decided to move all the figures involved with theoretical analysis to supplementary information part, in which, several models are presented for the shape of nuclei to justify our conclusions. The main text is kept concise with key experimental figures.

Fig. R1

Fig. R2

Fig. R3

Fig. R4

Aside from these fundamental points, some parts of the manuscript have been written (or rewritten) without enough care. Some sentences tell exactly the opposite of what one would think and there are inconsistencies at some places. For examples:

- In the abstract: “Nevertheless, the detailed mechanism of the transformation has yet been fully explored” likely means “Nevertheless, the detailed mechanism of the transformation has not yet been fully explored”

Authors’ response:

We have revised the sentence accordingly in the abstract.

- In Figure 6c and d, in the insets: “without 1 layer buffer” and “without 2 layer buffer” should likely be replaced by “with 1 layer buffer” and “with 2 layer buffer”

Authors’ response:

We have corrected the labels in the figures, which are now Figures S13.3c and Figure S13.3d.

- In Figure S15.2, the representations of the ZnO c-plane with O termination and that of the Zn termination are incorrect. The surface layer should contain only half of the O or Zn atoms, otherwise polarity is not compensated.

Authors’ response:

We agree with the reviewer’s comment. However, in order to avoid the misleading that might be caused, we have chosen to remove the representations of surface models. The calculation details have been mentioned in the text.

- In the caption of Figure S15-4, and line 485, saddle point configurations of Zn are found in (a-i) and those for O in (i-l) (instead of “saddle-point configurations for Zn and O adatoms (a-c) and (j-l) Zn and O adatoms”, which has no meaning.

Authors' response:

We have revised the figure caption accordingly (now Figure S13.2).

- In the same Figure, one can read that oxygen adsorption energy is equal to 2.253eV in configuration j, while, in the main text, it is said (line 228) that adsorption energies of oxygen atoms are larger than 4 eV

Authors' response:

We apologize for the confusion caused. In the main text, we meant to specify the adsorption energies corresponding to the most stable saddle positions. In the revised manuscript, we have chosen to use negative values for the adsorption energies (following the references: Phys. Rev. B 82, 155326 (2010)). We have also rectified the calculations of the adsorption energies following the formula used in the same reference and have updated the corresponding values in the figure (now Figure S13.2) and the plot (now Figure S13.3a). The descriptions in the main text and supplementary information have also been revised accordingly. We have added references for the formula and method used for the computation of adsorption energies and diffusion energy barriers.

I would like to say that this list is likely not exhaustive and the authors should check their manuscript more carefully. And they should correct their English or make it correct by someone with English or American mother tongue. Even many technical terms are incorrect (like “nucleation” used instead of “nucleus”, “Regoin” instead of “Region” in Figure 1, “volume energy per area” instead of cohesion energy per unit volume”, etc)

Authors' response:

We thank the reviewer for the careful proofreading. In the revised manuscript, we use “nucleation island” to represent “nucleus”, correct the word “Region” in Figure 1 and change the term “volume energy per area” to be “free energy change per volume” (following the standard definition in the references). We have also read through our manuscripts several times and tried our best to polish the English language.

To summarize, I think that, despite obvious improvements, the revised manuscript, both in its present form and content, is not yet ready for being accepted.

Authors' response:

We thank the reviewer for the detailed comments from two rounds of review, based on which, we have made extensive revisions to the theoretical part and improved our manuscript to a better shape. We hope that the revised manuscript has addressed the all the concerns from the reviewer.

Reviewer #3 (Remarks to the Author):

The authors have provided a detailed response to my concerns described in the original review, and I believe that the article is ready to publish in present form.

Authors' response:

We appreciate the reviewer's encouraging comments.

Reviewer #4 (Remarks to the Author):

In this manuscript, authors give the comprehensive characterization of the ZnO/MgO interface structures by STEM and XRD. I am not familiar with the theoretical calculation, so my comments will be mainly focused on the experimental part.

First I want to say that the quality of the STEM work is impressive. They belong to the best results we can get using present microscopes.

Although authors modified the title, but I still think the title is too broad. Why not specify the ZnO and MgO in the title?

Authors indexed the electron diffraction patterns in Fig. 3d as [011]ZnO and [110]MgO. The index for ZnO may be wrong. Based on the pattern, the incident electron beam direction should be along [1 -2 1 -3] direction in four-index, while [011] corresponding to [0 1 -1 1].

There are a lot of domains in both C-ZnO and m-ZnO, how large they are? Does the domain size relate to the deposition temperature? or interface strain?

Authors' response:

We thank the reviewer's encouraging comments on the high quality of our STEM work.

For the title, although the paper focuses on the interface between ZnO and MgO, we prefer to keep the broader title mentioning "hexagonal and cubic oxides", as one of the important points of the paper is to investigate the interface involving different crystalline phases; also, the paper aims to provide a guideline for the general case involving different crystalline phases.

For the index of the electron diffraction patterns, the relationship between Miller index (three-index) and Bravais-Miller (four-index) is:

For plane, $i = -(h+k)$, where (hkl) and (hkil) are Miller index and Bravais-Miller index, respectively;

For direction, if the Miller index is [uvw], and the corresponding Bravais-Miller index is [UVTW], then their relationship follows:

$$U = (2u-v)/3, V = (2v-u)/3, T = -(u+v)/3, W = w.$$

Therefore, in Fig. 3d, the Miller index [011] shall correspond to the Bravais-Miller index [-12-13], rather than [01-11]. Our original index should be correct.

For the rotational domains mentioned in our manuscript, they are originated from the mismatch of rotational symmetry at the interface and we have added a new session in the supplementary information (S4) to elaborate the related theoretical points. A typical domain size is about 10 nm width (as shown in Fig. S6a) and a length equivalent to the film thickness (about 150 nm). Whether the domain size depends on the deposition temperature or interface strain or other factors is a good question that we will explore in the future research work.

Reviewer #5 (Remarks to the Author):

I have reviewed the corrections to the paper and the authors' reply to the comments and I can recommend publication of the manuscript as is.

Authors' response:

We appreciate the reviewer's encouraging comments.

Reviewers' Comments:

Reviewer #4:

Remarks to the Author:

The authors clarified all my concerns in this new version. Therefore I have no objection to accept it for publication in the journal.

Reviewer #6:

Remarks to the Author:

The article "Interfaces between hexagonal and cubic oxides and their structure alternatives" studies the growth of ZnO on MgO. Dependent on oxygen partial pressure and temperature, different growth regimes are observed. I think that this paper would appeal to the readership of "Nature Communications" because it gives a lot of nicely measured information about the interfacial structure between crystals of different orientation. This Topic is of great interest for many technologic application. It is of fundamental scientific importance, too.

The work is supported by good simulations that support the main claims. I still hope that future calculations can give an even clearer picture of the interfacial structures - which is computationally complex. For an experimental paper, the computational support is more than enough.

In Response to Reviewer 1, the authors improved their model to a sufficient level (even giving a couple of alternative models if the situation is unclear -> all give the same results) and explained their computational assumptions in more detail. Finally, I agree with Reviewer 1 that the writing of the text could be better, but I think that the language quality of this version of the text is good enough for publication.

To sum up, the authors answered the comments of Reviewer 1 satisfactorily and the manuscript is suited for publication in Nature Communications in its present form.

We thank the reviewers for the encouraging and favorable comments for our manuscript.
Our detailed responses are listed below:

REVIEWERS' COMMENTS:

Reviewer #4 (Remarks to the Author):

The authors clarified all my concerns in this new version. Therefore I have no objection to accept it for publication in the journal.

Our response:

We are delighted for the approval from the reviewer.

Reviewer #6 (Remarks to the Author):

The article "Interfaces between hexagonal and cubic oxides and their structure alternatives" studies the growth of ZnO on MgO. Dependent on oxygen partial pressure and temperature, different growth regimes are observed. I think that this paper would appeal to the readership of "Nature Communications" because it gives a lot of nicely measured information about the interfacial structure between crystals of different orientation. This Topic is of great interest for many technologic application. It is of fundamental scientific importance, too.

The work is supported by good simulations that support the main claims. I still hope that future calculations can give an even clearer picture of the interfacial structures - which is computationally complex. For an experimental paper, the computational support is more than enough.

In Response to Reviewer 1, the authors improved their model to a sufficient level (even giving a couple of alternative models if the situation is unclear -> all give the same results) and explained their computational assumptions in more detail. Finally, I agree with Reviewer 1 that the writing of the text could be better, but I think that the language quality of this version of the text is good enough for publication.

To sum up, the authors answered the comments of Reviewer 1 satisfactorily and the manuscript is suited for publication in Nature Communications in its present form.

Our response:

We appreciate that the reviewer highlighted the points of our manuscript and recommended the publication in the present form. We also thank the reviewer's advice to further explore the complex interfacial structure, which we will plan for the future work.